# Synthetic Image Detection via Curvature of Diffusion Probability Flows

## Abstract

Synthetic image detection (SID) faces two major challenges: high computational cost from reconstruction-based methods and insufficient generalization. To address these issues, we propose a SID paradigm that leverages the ODE formulation of diffusion models. Instead of reconstructing images, our method analyzes probability flow trajectories from data distributions toward a Gaussian prior. We theoretically relate discrete step distances on the Wasserstein manifold to the kinetic energy of the probability flow. We further show empirically that trajectory deviation statistics derived from these distances correlate with reconstruction error and that real and synthetic images differ most in the early half of the diffusion inversion. In this regime, real images tend to exhibit higher curvature variance with occasional extreme deviations, whereas synthetic ones follow smoother and more consistent trajectories. Building on this observation, we introduce curvature features of probability flow trajectories as a discriminative signal for SID. To the best of our knowledge, this is the first work to exploit probability flow curvature for this task. Extensive experiments demonstrate that our method generalizes robustly to unseen models and achieves state of the art results across multiple benchmarks, while requiring less than half the FLOPs of reconstruction based detectors that perform full diffusion inversion.

## 1 Introduction

Continuous-time dynamic models (CTDMs) have emerged as a dominant class of generative models, evolving from energy-based and score-matching models (Hyvärinen & Dayan, 2005), through diffusion probabilistic models (Ho et al., 2020; Nichol & Dhariwal, 2021; Song et al., 2020a), Score-Based SDEs and Probability Flow ODEs (Song et al., 2020b; 2021; De Bortoli et al., 2021), to recent Flow Matching methods (Lipman et al., 2022; Liu et al., 2022; Albergo et al., 2023), achieving increasingly realistic image synthesis. However, the rapid proliferation of generative architectures has heightened concerns about the malicious use of synthetic media, motivating the need for detection frameworks that generalize across diverse and unseen generators.

Prior forensic methods (Corvi et al., 2023b; Ojha et al., 2023; Tan et al., 2024b; Sha et al., 2023; Liu et al., 2024) show strong performance on GAN-generated images but often fail to generalize to diffusion-based or newer generative models. Consequently, this work focuses on CTDMs. Among existing approaches, methods such as Wang et al. (2023a); Cazenavette et al. (2024); Ricker et al. (2024); Chu et al. (2025); Guillaro et al. (2025) introduced reconstruction error as a discriminative feature, yet most rely on replaying the full diffusion trajectory, without questioning whether the reconstruction error is already implicitly encoded in the probability flow.

In this paper, we adopt a unified ODE perspective and propose a detection paradigm that leverages the curvature of the probability flow velocity field as the primary discriminative feature (Fig. 1), complemented by diagonal high-frequency components extracted via wavelet transforms. Following Song et al. (2020a), a CTDM can be understood at the sample level as evolving a single data point along a probability flow ODE, which defines a continuous velocity field and maps noise distributions to data distributions. At the macroscopic level, this velocity field satisfies a continuity equation, describing how the probability density evolves smoothly over time. Integrating the reverse-time ODE then yields the instantaneous velocity at any intermediate time starting from a clean image.

We define the integration of the probability flow ODE over a data distribution as an *ODE pipeline*, which maps data distributions to a Gaussian prior. On the Wasserstein manifold, discrete step distances along this pipeline provide a kinetic energy upper bound on reconstruction error. We express this bound through a sum of non optimality terms over time steps, whose cumulative value measures how far a trajectory departs from the optimal transport path induced by the learned velocity field. Empirically, synthetic images have smaller cumulative non optimality than real ones. The gap is most pronounced in the first half of diffusion inversion and becomes negligible in the second half from the Gaussian back to the data distribution. This suggests that, in our setting, most of the reconstruction error is generated in the early part of the trajectory.

Building on this insight, we extract discriminative information from this early stage by computing curvature features of the probability flow trajectories. Real images show higher curvature variance and occasional extreme deviations, whereas synthetic images tend to follow smoother and more consistent paths. To the best of our knowledge, this is the first synthetic image detector that uses curvature features of probability flow trajectories. The method is trained on a single dataset yet remains robust across many recent generative models. It generalizes to a wide range of unseen models and uses less than half the FLOPs of reconstruction based detectors that perform full diffusion inversion. Compared to prior SOTA methods, it improves ACC by +10.6% and AUCROC by +8.2% across multiple benchmarks.

To summarize our contributions:

- We propose a new paradigm for synthetic image detection based on curvature features of ODE defined velocity fields.

- We analyze the ODE pipeline from an optimal transport view and find that synthetic images have lower total kinetic energy and follow lower energy transport paths than real images.

- We show that simple trajectory deviation statistics derived from the velocity field correlate strongly with reconstruction error, which links reconstruction based detectors to the geometry of the probability flow.

- We introduce a pseudo Gaussian curvature that compresses the temporal dimension of curvature features and improves their discriminative power.

- We complement curvature with diagonal high frequency wavelet components that capture fine local artifacts and further improve robustness and generalization.

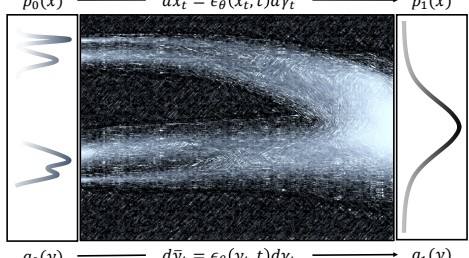

Figure 1: The figure shows the visualization of the velocity field, where brighter regions correspond to higher probability density. On the left are the initial distributions $p_0, q_0$ at time $t = 0$, which can be transformed into a Gaussian distribution by integrating the backward ODE.

## 2  RELATED WORK

**Artifact Detectors.**  Prior works have explored diverse strategies for synthetic image detection. CNN-based methods like Wang et al. (2020) train a ResNet-50 on ProGAN outputs with JPEG and blur augmentations. Frequency-based approaches, such as FrePGAN (Jeong et al., 2022) and FreqNet (Tan et al., 2024a), exploit high-frequency artifacts via model-specific analysis or FFT. UniFD (Ojha et al., 2023) decouples feature extraction and classification using a frozen CLIP encoder with a linear classifier. NPR (Tan et al., 2024b) targets autocorrelations induced by upsampling, while Fat-Former (Liu et al., 2024) combines semantic contrastive learning with wavelet-based artifact extraction. B-Free (Guillaro et al., 2025) constructs an unbiased dataset and employs a Vision Transformer to extract discriminative features.

**Reconstruction Error.**  DIRE: Wang et al. (2023a) propose Diffusion Reconstruction Error (DIRE), which differentiates real from DM-generated images by measuring reconstruction error. AEROB-LADE: Ricker et al. (2024) utilize autoencoder reconstruction error from latent DMs for a simple,

training-free approach. FakeInversion: Cazenavette et al. (2024) detect images generated by unseen text-to-image DMs using text-conditioned inversion. Luo et al. (2024) propose LaRE2, leveraging Latent Reconstruction Error (LaRE) with an Error-Guided Feature Refinement module for more distinct error feature extraction.

In this paper, we focus on and extend the second line of related work discussed above. While these approaches are all implemented as variants of reconstruction error, we question whether performing the entire reconstruction pipeline is truly necessary. The inversion process requires a full forward pass of the U-Net at every denoising or noising step, resulting in substantial computational and time overhead. Motivated by this limitation, we investigate whether certain inconsistency features can be directly captured within a single noising pass, thereby providing effective discriminative signals while avoiding the cost of full reconstruction.

## 3 BACKGROUND

### 3.1 ODE-BASED PROBABILITY FLOW AND VELOCITY FIELD

Let $p_0(x)$ denote the data distribution. In DDPM (Ho et al., 2020), the forward process is formulated as a stochastic differential equation (SDE) that gradually transforms $p_0(x)$ into a standard Gaussian distribution $p_T(x)$:

$$dx_t = f(x_t, t)dt + g(t)dw_t, \quad x_0 \sim p_0(x) \tag{1}$$

where $f : \mathbb{R}^D \to \mathbb{R}^D$ is the drift coefficient, $g(t) \in \mathbb{R}$ is the diffusion coefficient, and $w_t \in \mathbb{R}$ is a standard Wiener process. By sampling $x_T \sim p_T(x)$ and solving the reverse-time SDE, one can recover samples $x_0 \sim p_0(x)$:

$$dx_t = \big[f(x_t, t) - g(t)^2 \nabla_x \log p_t(x_t)\big]dt + g(t)d\bar{w}_t \tag{2}$$

Here, $\bar{w}_t$ denotes a standard Wiener process evolving backward from $T$ to 0. By setting the diffusion coefficient in the reverse process to zero, the stochastic trajectory becomes deterministic while preserving the same marginal distributions as in Eq. 2. This yields the Probability Flow ODE (PF-ODE) (Song et al., 2020b):

$$dx_t = \big[f(x_t, t) - \frac{1}{2}g(t)^2 \nabla_x \log p_t(x_t)\big]dt \tag{3}$$

Recent works (Lipman et al., 2022; Liu et al., 2022; Albergo et al., 2023) move beyond marginal distribution matching and instead directly learn a velocity field $\frac{dx_t}{dt} = v_\theta(x_t, t)$, which models the instantaneous velocity at each timestep to construct a continuous flow between distributions (illustrated in Figure 1). Under this view, the PF-ODE can be equivalently interpreted as modeling the velocity field using a score function:

$$v_\theta(x_t, t) = f(x_t, t) - \frac{1}{2}g(t)^2 \nabla_x \log p_t(x_t) \tag{4}$$

From this perspective, the velocity field $v_\theta$ satisfies the continuity equation:

$$\frac{\partial p_t(x)}{\partial t} + \nabla \cdot \big(v_\theta(x_t, t)\, p_t(x)\big) = 0 \tag{5}$$

### 3.2 OPTIMAL TRANSPORT AND W-DISTANCE

According to optimal transport theory, the cost of transporting one distribution $p(x)$ to another $q(x)$ is defined as:

$$C[p, q] = \inf_{\gamma \in \Pi(p,q)} \mathbb{E}_{(x,y)\sim\gamma}\big[c(x, y)\big] \tag{6}$$

where $\Pi(p, q)$ denotes the set of all couplings of $p(x)$ and $q(x)$, and $c(\cdot, \cdot)$ is the transport cost function. The Wasserstein-$\rho$ distance is then given by

$$W_\rho(p, q) = \left(\inf_{\gamma \in \Pi(p,q)} \mathbb{E}_{(x,y)\sim\gamma}\big[d(x, y)^\rho\big]\right)^{1/\rho} \tag{7}$$

where $d(x, y)$ is typically the Euclidean distance, and the cost is defined as its $\rho$-th power.

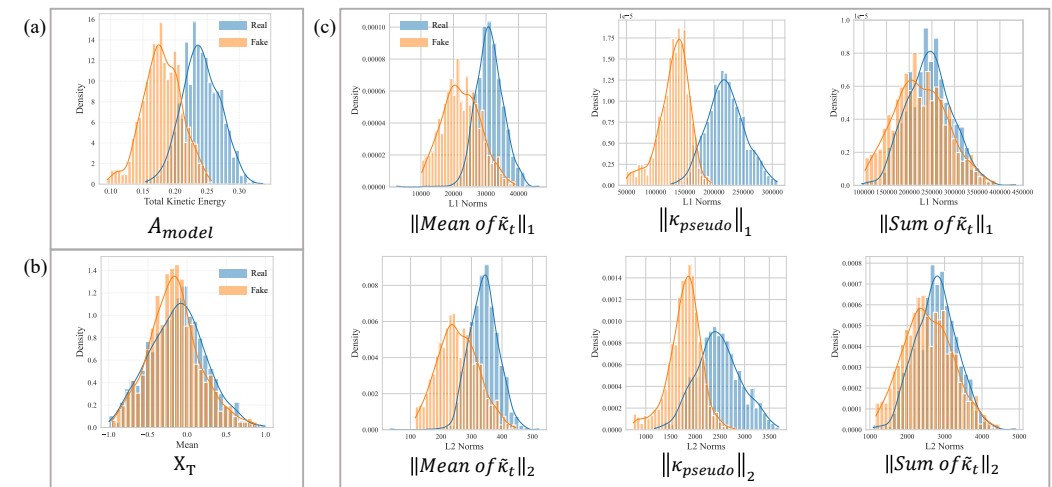

Figure 2: Both real and synthetic images (1,000 each) are randomly sampled from all datasets described in Section 6. The resulting histograms are: **(a)** total kinetic energy computed according to Theorem 2; **(b)** image means at time $T$, closely following a standard Gaussian distribution; and **(c)** curvature signal $\tilde{\kappa}_t$ under three different temporal compression strategies, highlighting differences in discriminative effectiveness.

The Wasserstein distance is a metric over probability distributions that preserves the geometry of the sample space, making it well suited for comparing distributions with partially non-overlapping supports. Under suitable conditions, the probability space endowed with the Wasserstein distance (the Wasserstein space) can be viewed as a Riemannian manifold, providing a natural setting to describe the continuous temporal evolution of probability densities.

From an optimal transport perspective, diffusion training corresponds to gradient descent on the KL divergence functional $\mathcal{F}[q] = KL(q\|p_{data})$ in the Wasserstein space. The neural network learns a velocity field $v_\theta(x_t, t)$ that approximates the steepest descent direction, yielding minimal-energy trajectories with smooth velocity fields whose endpoints are the generated samples.

These foundations highlight the close connection between PF-ODE formulations and optimal transport geometry. Building on this connection, we later show how the energy characteristics of probability flow trajectories reveal discriminative differences between real and synthetic images.

## 4 TRAJECTORY ANALYSIS IN WASSERSTEIN SPACE

### 4.1 VELOCITY FIELD AS $W_2$ DISTANCE ESTIMATION

We consider deterministic sampling trajectories induced by the PF-ODE corresponding to the marginal distributions of the forward VP-SDE. In particular, for the ADM model (Dhariwal & Nichol, 2021), the PF-ODE takes the form:

$$dx_t = \left[ -\tfrac{1}{2}\beta(t)x_t - \tfrac{1}{2}\beta(t)\nabla_x \log p_t(x_t) \right] dt \tag{8}$$

where the score function $\nabla_x \log p_t(x_t)$ is estimated via a noise prediction network $\epsilon_\theta(x_t, t)$:

$$\nabla_x \log p_t(x_t) \approx -\tfrac{1}{\sigma(t)} \epsilon_\theta(x_t, t) \tag{9}$$

with $\beta(t)$ are time-dependent constants, $\beta(t) = -\frac{d}{dt} log\bar{\alpha}_t$ and $\sigma(t)^2 = 1 - \bar{\alpha}_t$.

Solving Eq. 8 from $0$ to $T$ as an ODE pipeline, defines what we refer to as an ODE pipeline, which transports an initial distribution into an approximate Gaussian. As shown in Fig. 2(b), at terminal time $T$, both real and synthetic images are mapped close to a standard Gaussian. In the following, we use the term ODE pipeline with the default assumption that all trajectories are derived from the same CTDMs.

**Theorem 1.** *For an ODE pipeline applied to $x_0 \sim p_0$, the Wasserstein distance between two intermediate marginals is bounded by the mean kinetic energy at time $t$:*

$$W_2[p_t, p_{t+\Delta t}] \leq \Delta t \sqrt{\mathbb{E}_{x \sim p_t} \|v_\theta(x_t, t)\|^2}$$

**Theorem 2.** *Over the full pipeline from $0$ to $T$, the cumulative one-step $W_2$ distances are bounded by the total kinetic energy $A_{model}$:*

$$\sum_t W_2^2[p_t, p_{t+\Delta t}] \leq \delta t \int_0^{T\delta t} \mathbb{E}_{x \sim p_s} \|v_\theta(x_s, s)\|^2 ds$$

$$A_{model} := \int_0^{T\delta t} \mathbb{E}_{x \sim p_s} \|v_\theta(x_s, s)\|^2 ds$$

(Proofs are provided in Appendix A)

Theorems 1 and 2 formally characterize the transformation of marginals on the Wasserstein manifold under an ODE pipeline: Theorem 1 establishes a one-step bound, while Theorem 2 provides a cumulative bound that implicitly captures diffusion reconstruction error.

Previous work (Wang et al., 2023a; Ricker et al., 2024; Cazenavette et al., 2024; Luo et al., 2024; Chu et al., 2025) has shown that synthetic images yield lower reconstruction error. By Theorem 2, this corresponds to velocity fields with smaller kinetic energy. Thus, synthetic trajectories tend to be straighter than those of real images. To verify, we sample 1,000 real and 1,000 synthetic images (datasets in Sec. 6) and compute their total kinetic energy. As shown in Fig. 2(a), real images exhibit significantly larger energy, though the distributions overlap considerably, which motivates us to seek more effective statistical measures.

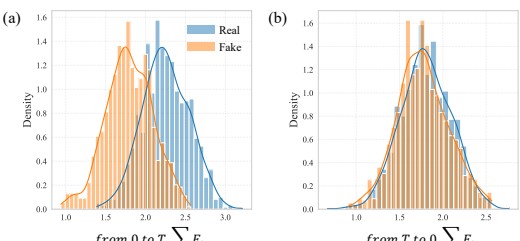

Figure 3: The data sampling is the same as in Fig. 2. The left panel corresponds to the first half of the diffusion reconstruction, and the right panel corresponds to the second half.

To further refine this analysis, we introduce a non optimality term at each step,

$$E_t = \Delta t \sqrt{\mathbb{E}_{x \sim p_t} \|v_\theta(x_t, t)\|^2} - W_2[p_t, p_{t+\Delta t}], \tag{10}$$

which is the difference between the stepwise kinetic energy bound in Theorem 1 and the actual one step Wasserstein distance. Summing over all time steps yields a cumulative non optimality $\sum E_t$. This quantity serves as a surrogate measure of how suboptimal a probability flow trajectory is relative to an energy minimizing transport path in Wasserstein space. Trajectories with smaller $\sum E_t$ can be interpreted as following transport paths that are more consistent with the minimal energy geometry induced by the learned velocity field.

Using the same data sampling protocol as in Fig. 2, Fig. 3 reports the cumulative non optimality across the forward and reverse diffusion directions. Although the ODE pipeline is theoretically bijective, we empirically observe an asymmetric behavior during reconstruction. Real image trajectories tend to drift into lower energy regions and end in $q_0$, while synthetic image trajectories start and end closer to $q_0$. This behavior can be summarized heuristically as mappings $p_0 \to \mathcal{N}(0, I) \to q_0$ for real images and $q_0 \to \mathcal{N}(0, I) \to q_0$ for synthetic ones. In particular, synthetic images consistently achieve smaller cumulative non optimality on our benchmarks, which empirically suggests that their trajectories tend to remain closer to the low energy transport paths preferred by the model. This observation also addresses our initial concern: in our experiments, the latter half of the trajectory, from $\mathcal{N}(0, I)$ back to $q_0$, contributes little discriminative information for distinguishing real and synthetic images and can therefore be discarded.

These findings suggest that the most informative differences between real and synthetic samples lie in how their trajectories bend and deviate from straight, low energy paths of the flow. From this perspective, synthetic images tend to follow smoother trajectories with fewer abrupt changes in velocity, whereas real images exhibit more irregular behavior. This motivates the use of curvature based descriptors that explicitly quantify trajectory bending.

## 4.2 ODE TAYLOR EXPANSION AND CURVATURE SURROGATE

Following the time reparameterization of Dockhorn et al. (2022), Eq. 8 becomes:

$$d\bar{x}_t = \epsilon_\theta(x_t, t)d\gamma_t \tag{11}$$

where $\gamma_t = \sqrt{\frac{1-\bar{\alpha}_t^2}{\bar{\alpha}_t}}$ , $\bar{x}_t = x_t\sqrt{1+\gamma_t^2}$ and $\epsilon_\theta(x_t, t) = -\sigma(t)\nabla_x \log p_t(x_t)$.

From the previous analysis, synthetic images tend to exhibit smoother velocity fields along this trajectory, which naturally leads us to consider curvature based measures of trajectory flatness. The geometric curvature in PF-ODE is defined as

$$\kappa(\gamma) = \frac{\|\bar{x}_\gamma'' - (\bar{x}_\gamma'' \cdot \hat{\epsilon}_\theta)\hat{\epsilon}_\theta\|}{\|\epsilon_\theta\|^2} \tag{12}$$

Additional details regarding Eq. 12 are provided in Appendix C.1, including the basic definition of curvature and the derivation of Eq. 12 under the PF-ODE setting. We approximate curvature via a second-order truncated Taylor method (TTM):

$$\bar{x}_{t_{n+1}} = \bar{x}_{t_n} + h_n\epsilon_\theta(x_{t_n}, t_n) + \frac{1}{2}h_n^2 \left.\frac{d\epsilon_\theta}{d\gamma_t}\right|_{(x_{t_n}, t_n)} \tag{13}$$

with step size $h_n = \gamma_{n+1} - \gamma_n$, The second-order term

$$\begin{aligned}\frac{d\epsilon_\theta(x_t,t)}{d\gamma_t} &= \frac{\partial\epsilon_\theta(x_t,t)}{\partial x_t}\frac{dx_t}{d\gamma_t} + \frac{\partial\epsilon_\theta(x_t,t)}{\partial t}\frac{dt}{d\gamma_t} \\ &= \frac{1}{\sqrt{\gamma_t^2+1}}\frac{\partial\epsilon_\theta(x_t,t)}{\partial x_t}\epsilon_\theta(x_t,t) - \frac{\gamma_t}{1+\gamma_t^2}\frac{\partial\epsilon_\theta(x_t,t)}{\partial x_t}x_t + \frac{\partial\epsilon_\theta(x_t,t)}{\partial t}\frac{dt}{d\gamma_t}\end{aligned} \tag{14}$$

involves two Jacobian-vector products (JVPs) $\frac{\partial\epsilon_\theta(x_t,t)}{\partial x_t}\epsilon_\theta(x_t,t)$, $\frac{\partial\epsilon_\theta(x_t,t)}{\partial x_t}x_t$, requiring additional backward passes and thus heavy computation.

To balance accuracy and efficiency, we adopt the correction term $\frac{1}{2}h_t^2\frac{d\epsilon_\theta}{d\gamma_t}$ as a surrogate curvature signal, and approximate it via finite differences:

$$\frac{1}{2}h_t^2\frac{d\epsilon_\theta}{d\gamma_t} \approx -\frac{\Delta\gamma_t^2\bar{\alpha}t^2\gamma_t\Delta\epsilon_{\theta_t}}{(1+\bar{\alpha}_t^2)\Delta\bar{\alpha}_t} := \tilde{\kappa}_t \tag{15}$$

This surrogate captures the dominant acceleration while avoiding costly JVPs, and empirically preserves trajectory characteristics (The derivation of Eq. 15 can be found in Appendix A.3).

Figure 2(c) shows histograms of $\tilde{\kappa}_t$. To enhance discriminability, we compare several temporal compression strategies, including sum, mean, and pseudo-Gaussian aggregation, each evaluated under both L1 and L2 norms. The results demonstrate that the $\kappa_{pseudo}$ with the L1 norm exhibits the least distributional overlap between real and synthetic images, thereby providing the strongest discriminative power (more details in Appendix C.2). This motivates us to adopt $\kappa_{pseudo}$ as the primary curvature descriptor in our subsequent analysis. Our use of a global temporal statistic over the PF-ODE trajectory is conceptually consistent with recent work that exploits temporal diffusion trajectories for origin attribution and related membership / model attribution tasks (Floros et al., 2024), although we focus on a different geometric quantity (curvature in Wasserstein space). Analogous to Gaussian curvature as the product of principal curvatures, we define a pseudo-Gaussian curvature $\kappa_{pseudo}$ as the product of the maximal and minimal trajectory curvatures observed along time:

$$\kappa_{pseudo} := \left(\max_{t\in[0,T]}(\tilde{\kappa}_t)\right) \cdot \left(\min_{t\in[0,T]}(\tilde{\kappa}_t)\right) \tag{16}$$

A formal justification of why this pseudo-Gaussian curvature effectively distinguishes real and synthetic distributions is provided in Appendix C.3.

## 5 METHOD

As shown in Figure 4, our full framework consists of two pipelines. Pipeline 1 is employed to extract curvature features. Since curvature is a second-order quantity and highly sensitive to variations in the input, we additionally use Pipeline 2 to extract zero-order information as an auxiliary signal.

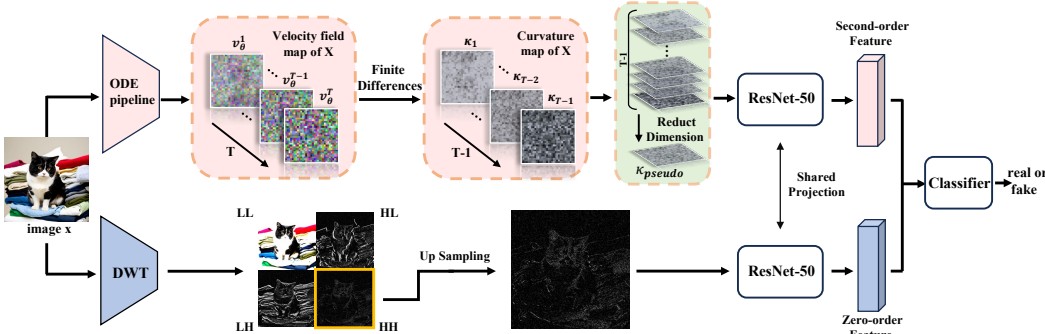

Figure 4: **Proposed method.** The model consists of two pipelines: one extracts second-order curvature features, and the other extracts zeroth-order diagonal high-frequency features. For the second-order features, we perform temporal dimensionality reduction using the proposed pseudo-Gaussian curvature. The two ResNet-50 backbones share the final projection layer to align features.

## 5.1 PIPELINE 1: CURVATURE FEATURE EXTRACTION.

Given a clean image $x_0$, we apply the ODE pipeline with a first-order Euler approximation to obtain intermediate states $x_t$. The corresponding first-order TTM from Eq. 13 reduces to:

$$\bar{x}_{t_{n+1}} = \bar{x}_{t_n} + h_n \epsilon_\theta(x_{t_n}, t_n) \tag{17}$$

At each time step, curvature features are computed using Eq. 15 and subsequently compressed along the temporal axis via the pseudo-Gaussian formulation in Eq. 16, yielding the pseudo-Gaussian curvature $\kappa_{pseudo}$. Although the L1 norm of $\kappa_{pseudo}$ already provides sufficient discriminative power for classification, to capture more comprehensive and fine-grained characteristics, we parameterize these features using a convolutional neural network. Therefore, $\kappa_{pseudo}$ is fed into a ResNet-50 backbone, which encodes the tensor into a compact curvature feature vector.

## 5.2 PIPELINE 2: IMAGE-BASED REPRESENTATION EXTRACTION.

Previous work (Corvi et al., 2023a; Tan et al., 2024a; Chu et al., 2025; Guillaro et al., 2025) has shown that synthetic images often contain frequency-domain artifacts. Based on these observations, we select the diagonal high-frequency components from the wavelet transform as zero-order information to complement the curvature-based features. Specifically, we employ a one-level discrete wavelet transform (DWT) using the Biorthogonal 1.3 ("bior1.3") basis with symmetric boundary extension. The DWT decomposes $x \in \mathbb{R}^{B \times C \times H \times W}$ into:

$$Y_\ell \in \mathbb{R}^{B \times C \times \frac{H}{2} \times \frac{W}{2}}, \tag{18}$$

$$\{Y_h^{(1)}, Y_h^{(2)}, Y_h^{(3)}\} \subset \mathbb{R}^{B \times C \times \frac{H}{2} \times \frac{W}{2}}. \tag{19}$$

where $Y_\ell$ is the low-frequency approximation and $\{Y_h^{(1)}, Y_h^{(2)}, Y_h^{(3)}\}$ are the horizontal (LH), vertical (HL), and diagonal (HH) detail subbands, respectively.

We select the diagonal detail coefficients $Y_h^{(3)}$, which capture edge and texture variations along oblique orientations. To align this feature map with the original spatial resolution, we upsample via bilinear interpolation:

$$\tilde{Y}_h^{(3)} = \text{Interp}\left(Y_h^{(3)}; [H, W]\right) \tag{20}$$

The resulting output $\tilde{Y}_h^{(3)}$, now spatially aligned with the original image, are then encoded by a ResNet-50 to produce hidden representations.

## 5.3 PROJECTION AND CLASSIFICATION.

In both pipelines, the final fully connected layers of the ResNet-50 networks project their outputs into a shared 512-dimensional subspace, aligning curvature-based and frequency-domain features.

| Eval Set | UFD | FakeInv. | NPR | FatFormer | B-Free | ours | TPR@ 5%FPR |
|---|---|---|---|---|---|---|---|
| DALL·E 2 | 0.700 / 0.776 | 0.678 / 0.747 | 0.945 / 0.995 | **0.987 / 0.998** | 0.906 / 0.969 | 0.851 / 0.953 | 0.762 |
| DALL·E 3 | 0.473 / 0.480 | 0.698 / 0.759 | 0.720 / 0.807 | 0.801 / 0.881 | **0.912 / 0.972** | 0.860 / 0.961 | 0.759 |
| Midjourney v5/6 | 0.558 / 0.592 | 0.606 / 0.664 | 0.778 / 0.854 | 0.560 / 0.627 | 0.946 / 0.988 | **0.965 / 0.993** | 0.966 |
| Imagen | 0.538 / 0.575 | 0.720 / 0.807 | **0.982 / 0.998** | 0.946 / 0.995 | 0.908 / 0.970 | 0.960 / 0.991 | 0.983 |
| Kandinsky 2 | 0.541/ 0.562 | 0.652 / 0.699 | 0.737 / 0.811 | 0.676 / 0.700 | 0.778 / 0.860 | **0.950 / 0.995** | 0.979 |
| Kandinsky 3 | 0.600 / 0.637 | 0.684 / 0.743 | 0.749 / 0.786 | 0.694 / 0.749 | 0.801 / 0.884 | **0.948 / 0.991** | 0.980 |
| PixArt-$\alpha$ | 0.606 / 0.647 | 0.669 / 0.730 | 0.785 / 0.860 | 0.671 / 0.791 | 0.830 / 0.911 | **0.974 / 0.997** | 0.982 |
| Playground 2.5 | 0.562 / 0.587 | 0.591 / 0.625 | 0.725 / 0.843 | 0.729 / 0.803 | 0.796 / 0.879 | **0.863 / 0.899** | 0.810 |
| SDXL-DPO | 0.647 / 0.702 | 0.801 / 0.881 | 0.785 / 0.901 | 0.826 / 0.913 | 0.647 / 0.759 | **0.843 / 0.957** | 0.776 |
| SDXL | 0.620 / 0.663 | 0.737 / 0.807 | 0.778 / 0.860 | 0.651 / 0.656 | 0.651 / 0.776 | **0.867 / 0.962** | 0.798 |
| Seg-MOE | 0.586 / 0.620 | 0.664 / 0.713 | 0.732 / 0.801 | 0.830 / 0.899 | 0.705 / 0.777 | **0.963 / 0.995** | 0.978 |
| SSD-1B | 0.585 / 0.628 | 0.724 / 0.794 | 0.753 / 0.872 | 0.859 / 0.947 | 0.833 / 0.919 | **0.967 / 0.996** | 0.984 |
| Stable-Cascade | 0.633 / 0.682 | 0.694 / 0.749 | 0.744 / 0.897 | 0.538 / 0.766 | 0.824 / 0.906 | **0.963 / 0.996** | 0.981 |
| Segmind Vega | 0.587 / 0.623 | 0.733 / 0.811 | 0.774 / 0.862 | 0.812 / 0.936 | 0.819 / 0.901 | **0.937 / 0.983** | 0.965 |
| Würstchen 2 | 0.640 / 0.697 | 0.658 / 0.705 | 0.761 / 0.805 | 0.730 / 0.858 | 0.807 / 0.890 | **0.871 / 0.916** | 0.877 |
| ADM | 0.682 / 0.779 | 0.676 / 0.700 | 0.697 / 0.746 | 0.784 / 0.917 | 0.725 / 0.843 | **0.998 / 1.000** | 0.999 |
| Glide | 0.640 / 0.685 | 0.749 / 0.786 | 0.784 / 0.857 | 0.879 / 0.959 | 0.700 / 0.739 | **0.982 / 0.998** | 0.990 |
| VQDM | 0.840 / 0.876 | 0.648 / 0.681 | 0.781 / 0.812 | 0.868 / 0.969 | 0.885 / 0.928 | **0.945 / 0.995** | 0.976 |
| FLUX | 0.599 / 0.637 | 0.651 / 0.656 | 0.795 / 0.887 | 0.544 / 0.618 | 0.862 / 0.900 | **0.963 / 0.996** | 0.983 |
| Stable Diffusion 1.4 | 0.651 / 0.663 | 0.597 / 0.612 | 0.932 / 0.954 | 0.767 / 0.861 | 0.997 / 1.000 | **0.999 / 1.000** | 0.999 |
| Stable Diffusion 1.5 | 0.647 / 0.684 | 0.639 / 0.675 | 0.946 / 0.966 | 0.780 / 0.890 | 0.995 / 0.997 | **0.999 / 1.000** | 0.999 |
| Stable Diffusion 3 | 0.613 / 0.638 | 0.600 / 0.646 | 0.789 / 0.875 | 0.583 / 0.691 | 0.996 / 0.997 | **0.998 / 0.999** | 0.999 |
| **Average** | 0.616 / 0.656 | 0.676 / 0.727 | 0.794 / 0.866 | 0.751 / 0.837 | 0.833 / 0.899 | **0.939 / 0.981** | 0.933 |

Table 1: Results (**ACC / AUCROC**) of several SOTA methods evaluated on our collected diffusion models and updated generative models. All the modles are re-trained with the official codes. Due to space constraints, we report the results of Methods CNNDet (Wang et al., 2020) and DMDet (Corvi et al., 2023b) in Appendix F (Table 5).

| NFEs | trained on | | | Average |
|---|---|---|---|---|
| | SD + LAION | ProGAN + LSUN | ADM + LAION | |
| 5 | 0.876 / 0.912 | 0.885 / 0.928 | 0.890 / 0.933 | 0.883 / 0.924 |
| 10 | 0.939 / 0.981 | 0.894 / 0.946 | 0.927 / 0.971 | 0.920 / 0.966 |
| 15 | 0.934 / 0.979 | 0.892 / 0.945 | 0.922 / 0.969 | 0.916 / 0.964 |
| 20 | 0.919 / 0.956 | 0.874 / 0.919 | 0.897 / 0.959 | 0.897 / 0.945 |
| 50 | 0.910 / 0.948 | 0.875 / 0.923 | 0.898 / 0.959 | 0.894 / 0.943 |

Table 2: Ablation results ACC / AUCROC on different diffusion steps and different training dataset.

| Only | ACC / AUCROC |
|---|---|
| Cur. | 0.803 / 0.928 |
| DWT (Yl) | 0.637 / 0.687 |
| DWT (LH) | 0.707 / 0.803 |
| DWT (HL) | 0.779 / 0.864 |
| DWT (HH) | 0.796 / 0.902 |

Table 3: Ablation results ACC / AUCROC on curvature only or DWT only.

These two 512-d vectors are concatenated and passed to the final classification layer. Notably, the diagonal high-frequency band (HH subband) is particularly sensitive to oblique variations and irregular details, capturing fine-grained, directionally structured textures that standard convolutions often overlook. By integrating these complementary modalities, our framework assesses how frequency-domain artifacts correlate with less-flat ODE trajectories, enabling cross-validation of cues and substantially improving robustness and generalization.

# 6 EXPERIMENTS

**Datasets.** For some evaluation data, such as those from Imagen (Saharia et al., 2022), Midjourney (Midjourney, 2022), and DALL·E 3, are obtained via Hugging Face using KPI. Additionally, we generate thousands of high-fidelity images using open-source text-to-image models based on COCO (Lin et al., 2014) prompts. All settings are kept consistent with those in FakeInversion. The evaluation datasets includes fake images from Kandinsky 2 (Shakhmatov et al., 2023), Kandinsky 3 (Arkhipkin et al., 2023), PixArt (Chen et al., 2023), SDXL-DPO (Wallace et al., 2024), SDXL (Podell et al., 2023), SegMoE (Harish et al., 2024), SSD-1B (Gupta et al., 2024), Stable-Cascade (Pablo et al., 2023), Segmind-Vega (Gupta et al., 2024), Würstchen 2 (Pablo et al., 2023), ADM (Dhariwal & Nichol, 2021), GLIDE (Nichol et al., 2021), VQDM (Gu et al., 2022), FLUX (Labs., 2024), Stable Diffusion 1.4 / 1.5 (Rombach et al., 2022) and Stable Diffusion 3 (Esser et al., 2024). Further details about the datasets are provided in Appendix D.

**Metrics.** We report detection ACC, AUC-ROC as the primary metrics, and additionally provide TPR at 5% FPR as a supplementary measure.

**Baselines.** We use recent methods with publicly available code and pretrained weights as our baselines: CNNDet (Wang et al., 2020), DMDet (Corvi et al., 2023b), UniFD (Ojha et al., 2023), FakeInversion (Cazenavette et al., 2024), NPR (Tan et al., 2024b), FatFormer (Liu et al., 2024), B-Free (Guillaro et al., 2025). For further implementation details of our model, please refer to Appendix E.

| Method | Ref. | #Params | #FLOPs |
|--------|------|---------|--------|
| UFD | CVPR 2023 | 427.6 M | 77.8 B |
| NPR | CVPR 2024 | 1.4 M | 2.3 B |
| FatFormer | CVPR 2024 | 577.3 M | 128.0 B |
| B-Free | CVPR 2025 | 85.5 M | 553.1 B |
| FakeInv. | CVPR 2024 | 5.2 B | 10385.0 B |
| LaRE$^2$ | CVPR 2024 | 479.5 M | 41405.3 B |
| AEROBLADE | CVPR 2024 | 84.0 M | 446.0 B |
| FIRE | CVPR 2025 | 484.0 M | 3346.2 B |
| ours | - | 298.4M | 518.4 B |

Table 4: We compare the number of model **parameters** and **FLOPs**. The upper part are non–reconstruction approaches, the lower part are reconstruction–based and the last one is our PF-ODE based method.

## 6.1 MAIN RESULTS

Table 1 presents a comprehensive comparison across a broad range of recent generative models. Our method consistently achieves superior performance on nearly all benchmark datasets, with an average ACC/AUCROC of 0.939 / 0.981, substantially outperforming SOTA baselines (B-Free: 0.833 / 0.899). For instance, on high-fidelity diffusion models such as ADM, Glide, and Stable Diffusion 1.4/1.5/3, our framework reaches near-perfect detection (ACC 0.998 to 0.999, AUCROC 0.998 to 1.000), while baseline methods show noticeable gaps (e.g., B-Free ACC 0.700 to 0.997). This demonstrates the efficiency of our approach in capturing the distinctive generative characteristics of a wide spectrum of diffusion models, including text-to-image systems such as Midjourney v5/6 and Imagen. Unlike conventional methods that rely on model-specific features or complex reconstruction pipelines, our framework generalizes well without requiring text prompts or additional large models. By leveraging curvature-based features derived from the velocity field of the underlying differential equations, together with high-frequency components extracted via wavelet transforms, our approach effectively detects subtle inconsistencies introduced during generation.

Additionally, Table 1 reports the true positive rate (TPR) at a fixed false positive rate (FPR), which is particularly informative for practical deployment (set to 5% in this study). Our TPR values are consistently high across all datasets, with an average of 0.933. Although some models show slight variations (e.g., Playground 2.5: TPR 0.810), the overall performance remains balanced, highlighting the robustness of the proposed framework.

**Ablation of Curvature and Wavelet.** Table 3 reports an ablation study over different wavelet subbands and the two feature branches. Among all subbands, the diagonal HH subband achieves the highest accuracy and is therefore used in our wavelet branch. Using only the curvature branch yields 80.3% ACC, while using only the wavelet branch with the HH subband gives 79.6% ACC. Both configurations are clearly weaker than the full model, which reaches 93.9% ACC when the two branches are fused. This gap indicates that curvature and diagonal high frequency features capture largely complementary evidence rather than redundant cues.

Curvature features encode the temporal smoothness of probability flow trajectories and are therefore sensitive to how easily a sample is transported toward the Gaussian prior. However, They are relatively insensitive to fine spatial artifacts. In contrast, the HH subband emphasizes diagonal edges and fine textures in the image space, which respond strongly to local synthesis artifacts and compression patterns but do not reflect the global transport geometry of the probability flow. Combining the two branches allows the classifier to jointly exploit global dynamical cues and local high frequency artifacts, which substantially improve the accuracy.

**Ablation of NFEs and Training set.** We studied the effects of dataset selection and diffusion steps on model performance. As shown in Table 2, SD+LAION achieves the best training results. Although ProGAN+LSUN lags behind, it still outperforms the baseline methods. GAN based models can also be interpreted as diffusion processes along the temporal dimension from a gradient flow perspective (Yi et al., 2023), but their stronger model specific characteristics may compromise cross model generalization in our setting.

Regarding the number of function evaluations (NFEs), 10 and 15 steps yield the best results and show nearly identical performance. With only 5 NFEs, each ODE step spans a large time interval. This leads to a coarse approximation of the probability flow and degrades both distance and curvature estimates. This is expected in our setting because the chosen ODE pipeline does not support one step generation. From 10 NFEs onwards, the discretization is already fine enough to capture the relevant geometric differences in the trajectories, so further increasing NFEs brings only diminishing returns. Beyond 20 NFEs, performance changes are minor while the computational cost continues to grow, which explains the plateau observed in Table 2.

**Complexity and Efficiency.** Table 4 reports parameter counts and FLOPs for our method and several strong baselines. The upper part lists non-reconstruction methods, and the lower part lists reconstruction-based ones. NPR (Tan et al., 2024b) is an extremely lightweight approach with the lowest computational cost among all methods. By contrast, reconstruction-based methods that replay the full diffusion process are much more expensive: FakeInversion, LaRE$^2$ and FIRE require about $1.0 \times 10^4$B, $4.1 \times 10^4$B and $3.3 \times 10^3$B FLOPs per image, respectively. Our PF ODE based detector also relies on the diffusion model but needs only 518.4B FLOPs, which is less than half the cost of these full inversion approaches and comparable to AEROBLADE and the current state of the art B-Free. We currently use a basic PF ODE solver, and combining this pipeline with accelerated samplers such as DPM Solver, AMED Solver or consistency models could further reduce the required NFEs and FLOPs.

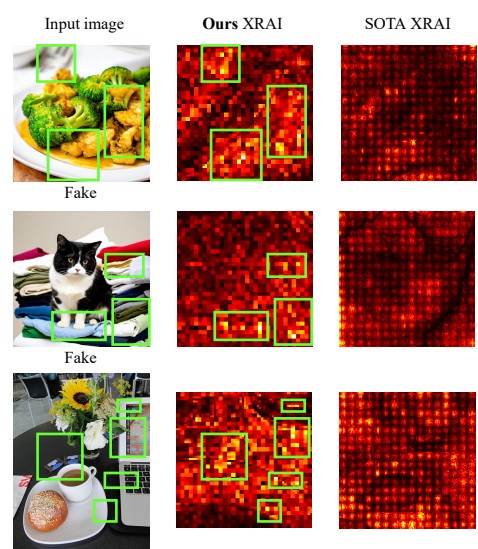

Figure 5: The green boxes highlight the most salient regions identified by our model. We visualize these regions using the post-hoc explainability method XRAI (Kapishnikov et al., 2019). Areas in synthetic images with incorrect lighting or perspective exhibit the highest saliency in our model.

**Saliency Analysis.** Figure 5 visualizes the most salient regions identified by our model using XRAI. For synthetic images, the saliency tends to concentrate on localized structural details, while real images exhibit more broadly distributed evidence supporting authenticity. This suggests that our detector relies on fine-grained, region-specific cues rather than purely global statistics, indicating potential for localized anomaly detection and artifact correction. In contrast, the baseline B-Free mainly emphasizes global weighted aggregation and depends on large-scale training (360k images: 51k real, 309k fake) with ViT backbones, whereas our model achieves strong results using only 80k images (40k real, 40k fake).

## 7 CONCLUSIONS

We present a novel framework for synthetic image detection that moves beyond traditional reconstruction based methods by leveraging curvature features derived from the velocity field of diffusion models. By analyzing reconstruction error from an optimal transport perspective, we relate discrete Wasserstein step distances and non optimality statistics to trajectory deviation. On our benchmarks, synthetic images tend to follow lower energy trajectories that stay closer to low energy transport paths than those of real images. We summarize these differences using pseudo Gaussian curvature features. These curvature features, combined with high frequency components extracted via discrete wavelet transforms as a zeroth order complement, enable the model to capture subtle generative artifacts with high discriminative power. Our approach improves generalization across diverse diffusion models and reduces computational overhead compared to full reconstruction pipelines. We hope that this work will inspire further research in multimedia forensics and foster progress in the community.

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

# Appendix

## A  PROOF

### A.1  THEOREM 1

**Theorem 1.** *When the ODE pipeline is applied to an initial distribution $x_0 \sim p_0$, the Wasserstein distance between the two marginal distributions at any intermediate times $t$ and $t + \Delta t$ is controlled by the mean kinetic energy at time $t$:*

$$W_2[p_t, p_{t+\Delta t}] \leq \Delta t \sqrt{\mathbb{E}_{x \sim p_t} \|v_\theta(x_t, t)\|^2}$$

*Proof.* From the definition of the $W_2$ distance in Eq.7, we have:

$$W_2[p_t, p_{t+\Delta t}] \leq \sqrt{\mathbb{E}_{x \sim p_t} \|x_{t+\Delta t} - x_t\|^2} \tag{21}$$

For the discrete ODE pipeline using the explicit Euler update:

$$x_{t+\Delta t} = x_t + v_\theta(x_t, t)\Delta t \tag{22}$$

Substituting this into the previous bound yields:

$$W_2[p_t, p_{t+\Delta t}] \leq \sqrt{\mathbb{E}_{x \sim p_t} \|v_\theta(x_t, t)\Delta t\|^2} \tag{23}$$

$$= \Delta t \sqrt{\mathbb{E}_{x \sim p_t} \|v_\theta(x_t, t)\|^2} \tag{24}$$

$\square$

### A.2  THEOREM 2

**Theorem 2.** *From time $0$ to $T$, the cumulative one-step $W_2$ distances over the ODE pipeline is controlled by the total kinetic energy $A_{model}$:*

$$\sum_t W_2^2[p_t, p_{t+\Delta t}] \leq \delta t \int_0^{T\delta t} \mathbb{E}_{x \sim p_s} \|v_\theta(x_s, s)\|^2 ds$$

$$A_{model} := \int_0^{T\delta t} \mathbb{E}_{x \sim p_s} \|v_\theta(x_s, s)\|^2 ds$$

*Proof.* According to Theorem 1, squaring the one-step Wasserstein distance and then summing over the entire time horizon of the ODE pipeline yields:

$$\sum_{t=0}^{T-1} W_2^2[p_t, p_{t+\Delta t}] \leq \sum_{t=0}^{T-1} \Delta t^2 \mathbb{E}_{x \sim p_t} \|v_\theta(x_t, t)\|^2 \tag{25}$$

For any $t \in [0, (T-1)\delta t]$, by applying time rescaling and the Benamou–Brenier inequality, the discrete summation can be mapped to its continuous-time counterpart:

$$\sum_{t=0}^{T-1} W_2^2[p_t, p_{t+\Delta t}] \leq \sum_{t=0}^{T-1} \delta t \int_t^{t+\delta t} \int \|v_\theta(x_s, s)\|^2 dp_s ds$$

$$= \delta t \int_0^{T\delta t} \int \|v_\theta(x_s, s)\|^2 dp_s ds$$

$$= \delta t \int_0^{T\delta t} \mathbb{E}_{x \sim p_s} \|v_\theta(x_s, s)\|^2 ds \tag{26}$$

In this formulation, we define the integral term as the total kinetic energy of the velocity field:

$$A_{model} := \int_0^{T\delta t} \mathbb{E}_{x \sim p_s} \|v_\theta(x_s, s)\|^2 ds \tag{27}$$

$\square$

## A.3 Correction Term of the Second-Order TTM

For the second-order correction term $\frac{1}{2}h_t^2\frac{d\epsilon_\theta}{d\gamma_t}$, we first differentiate the numerator and denominator with respect to $t$:

$$\frac{1}{2}h_t^2\frac{d\epsilon_\theta}{d\gamma_t} = \frac{1}{2}h_t^2\frac{d\epsilon_\theta/dt}{d\gamma_t/dt} \tag{28}$$

Then, we apply a simple finite difference scheme to compute the numerator:

$$\frac{d\epsilon_\theta(x_t,t)}{dt} = \frac{\epsilon_\theta(x_{t+\Delta t}, t+\Delta t) - \epsilon_\theta(x_t,t)}{\Delta t} \tag{29}$$

Given that $\gamma_t = \sqrt{\frac{1-\bar{\alpha}_t^2}{\bar{\alpha}_t}} = \sqrt{\frac{1}{\bar{\alpha}_t} - \bar{\alpha}_t}$, differentiating with respect to $t$ yields:

$$\begin{aligned}
\frac{d\gamma_t}{dt} &= \frac{1}{2}\left(\frac{1}{\bar{\alpha}_t} - \bar{\alpha}_t\right)^{-\frac{1}{2}} \cdot \frac{d}{dt}\left(\frac{1}{\bar{\alpha}_t} - \bar{\alpha}_t\right) \\
&= \frac{1}{2}\left(\frac{1}{\bar{\alpha}_t} - \bar{\alpha}_t\right)^{-\frac{1}{2}}\left(-\frac{1}{\bar{\alpha}_t^2}\frac{d\bar{\alpha}_t}{dt} - \frac{d\bar{\alpha}_t}{dt}\right) \\
&= -\frac{1}{2}\left(\frac{1}{\bar{\alpha}_t} - \bar{\alpha}_t\right)^{-\frac{1}{2}}\left(\frac{1+\bar{\alpha}_t^2}{\bar{\alpha}_t^2}\frac{d\bar{\alpha}_t}{dt}\right) \\
&= -\frac{1+\bar{\alpha}_t^2}{2\bar{\alpha}_t^2\gamma_t}\frac{d\bar{\alpha}_t}{dt}
\end{aligned} \tag{30}$$

Since $\bar{\alpha}_t$ is a predefined set of hyperparameters, its derivative with respect to $t$ can also be approximated using finite differences. Then, substituting the above into Eq. 28, we obtain:

$$\frac{1}{2}h_t^2\frac{d\epsilon_\theta}{d\gamma_t} \approx -\frac{\Delta\gamma_t^2\bar{\alpha}_t^2\gamma_t\Delta\epsilon_{\theta_t}}{(1+\bar{\alpha}_t^2)\Delta\bar{\alpha}_t} \tag{31}$$

# B   COMPUTE $E_t$

In Eq. 10 we define the non-optimality term $E_t$. This section details the algorithm for computing its cumulative sum $\sum_t E_t$.

For clarity of notation, we denote the root-mean-square norm of the model-predicted velocity field under the current distribution $p_t$ in Eq. 10 as $K_t$:

$$\sqrt{\mathbb{E}_{x\sim p_t}\|v_\theta(x_t,t)\|^2} \approx \sqrt{\frac{1}{N}\sum_{i=1}^N \|v_\theta(x_t^i,t)\|^2} := K_t \tag{32}$$

Accordingly, Eq. 10 can be rewritten in a simplified form as:

$$E_t = \Delta t \cdot K_t - W_2[p_t, p_{t+\Delta t}] \tag{33}$$

Based on this, the cumulative sum is computed as

$$\mathcal{E}_{0:T-1} = \sum_{t=0}^{T-1} E_t \tag{34}$$

This metric characterizes the discrepancy between the model-provided upper bound on kinetic energy and the true Wasserstein distance, reflecting the degree of non-optimality of the overall velocity field relative to the optimal transport.

In our experiments, we approximate the $W_2$ distance between consecutive time-step distributions using an entropy-regularized Sinkhorn (Cuturi, 2013) optimal transport algorithm. The implementation employs the GeomLoss (SamplesLoss) interface with the cost function exponent set to $p = 2$, corresponding to the $W_2$ distance. To maintain numerical stability in high-dimensional sample spaces, we set the blur parameter to 0.01 and the multi-scale scaling factor to 0.9. When iterating on the GPU, these parameters strike a balance between accuracy and computational efficiency. We provide pseudocode in Algorithm 1.

---

**Algorithm 1** Cumulative Non-Optimality Computation

---

**Require:**
Samples $\{x_t^i\}$ for times $t = 0, \ldots, T$
Model velocity field $v_\theta(x, t)$
Time step $\Delta t$
Sinkhorn parameters: $p = 2$, blur $= 0.01$, scaling $= 0.9$
**Ensure:** $\{W2_t\}, \{E_t\}$, cumulative error $\mathcal{E}$
1: Initialize $\mathcal{E} \leftarrow 0$
2: **for** $t = 0$ **to** $T - 1$ **do**
3:     $K_t \leftarrow \sqrt{\frac{1}{N} \sum_i \|v_\theta(x_t^i, t)\|^2}$
4:     $W2sq_t \leftarrow \text{Sinkhorn}(x_t, x_{t+1}; p, \text{blur}, \text{scaling}), W2_t \leftarrow \sqrt{W2sq_t}$
5:     Compute non-optimality term: $E_t \leftarrow \Delta t \cdot K_t - W2_t$
6:     Update cumulative error: $\mathcal{E} \leftarrow \mathcal{E} + E_t$
7: **end for**
8: **return** $\mathcal{E}$

---

## C    CURVATURE FEATURES

### C.1    GEOMETRIC CURVATURE OF PF-ODE TRAJECTORIES

We briefly recall the geometric definition of curvature and show how Eq. 12 arises in our setting. Recall the shared PF–ODE:

$$\frac{dx_t}{dt} = v(x_t, t), \quad x_0 \sim r_0 \tag{35}$$

where $r_0 \in \{p_0, q_0\}$ denotes the real or synthetic distribution.

#### C.1.1    GEOMETRIC CURVATURE IN $\mathbb{R}^D$.

The geometric curvature is defined as the norm of the normal component of the acceleration, normalized by the squared speed:

$$\kappa(t) \;=\; \frac{\left\| P_{v(t)^\perp} \, v'(t) \right\|}{\|v(t)\|^2} \tag{36}$$

where $v'(t) = \frac{dv}{dt}(t)$, and $P_{v(t)^\perp}$ denotes the orthogonal projection onto the subspace orthogonal to $v(t)$. Since

$$P_{v(t)^\perp} \, v'(t) = v'(t) - \big(v'(t) \cdot \hat{v}(t)\big)\hat{v}(t) \tag{37}$$

with $\hat{v}(t) := \frac{v(t)}{\|v(t)\|}$, Eq. 36 can be written equivalently as

$$\kappa(t) \;=\; \frac{\left\| v'(t) - \big(v'(t) \cdot \hat{v}(t)\big)\hat{v}(t) \right\|}{\|v(t)\|^2} \tag{38}$$

#### C.1.2    SPECIALIZATION TO REPARAMETERIZATION PF–ODE TRAJECTORIES.

After the time reparameterization of Eq. 11, identifying $x(t)$ in Eq. 38 with the reparameterized trajectory $\bar{x}_t(\gamma_t)$ and using

$$v(\gamma_t) = \frac{d\bar{x}}{d\gamma_t} = \epsilon_\theta(x_t, t), \quad v'(\gamma_t) = \frac{d\,\epsilon_\theta(x_t, t)}{d\gamma_t} = \frac{d^2 \bar{x}_t}{d\gamma_t^2} \tag{39}$$

we obtain the curvature along the PF–ODE trajectory:

$$\kappa(\gamma_t) = \frac{\left\| \frac{d^2 \bar{x}_t}{d\gamma_t^2} - \big(\frac{d^2 \bar{x}_t}{d\gamma_t^2} \cdot \hat{\epsilon}_\theta(x_t, t)\big)\hat{\epsilon}_\theta(x_t, t) \right\|}{\|\epsilon_\theta(x_t, t)\|^2} \tag{40}$$

It can therefore be written in a simplified form as:

$$\kappa(\gamma) = \frac{\|\bar{x}_\gamma'' - (\bar{x}_\gamma'' \cdot \hat{\epsilon}_\theta)\hat{\epsilon}_\theta\|}{\|\epsilon_\theta\|^2} \tag{41}$$

which is exactly the expression used in Eq. 12 of the main text. Eq. 12 provides a concise geometric expression for curvature. Although written in an abstract form, it becomes fully operational once its only nontrivial component, the second–order trajectory derivative $\bar{x}_\gamma'' = \frac{d\epsilon_\theta(x_t,t)}{d\gamma_t}$, is made explicit. This quantity follows directly from the reparameterized PF–ODE via the chain rule, as shown in Eq. 14 in the main text.

We emphasize that expanding Eq. 12 into fully explicit coordinates is possible but yields a lengthy formula with no additional conceptual value. The expression above already provides the complete ODE form of curvature for the reparameterized PF–ODE, and all quantities appearing in Eq. 12 are directly computable from the model.

## C.2 COMPRESSING CURVATURE FEATURES ALONG TIME

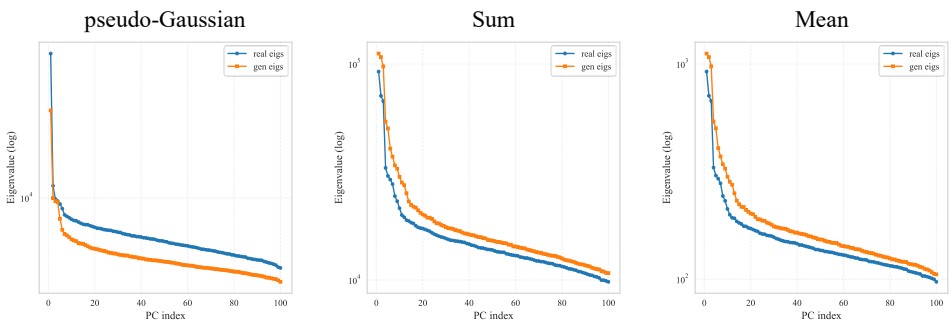

Figure 6: Eigenvalue curves from randomized SVD under different temporal compression methods.

In Fig. 2, we present histograms of curvature features obtained using three different methods for compressing the temporal dimension. From the histograms, one can already discern which method is more discriminative. Here, we further provide principal component analysis (PCA) of these different methods.

We perform randomized SVD on the curvature features computed via temporal summation, mean, and pseudo-Gaussian methods for both real and synthetic images, and obtain the top eigenvalues for each principal component.

As shown in Fig. 6, we compare the top eigenvalue curves obtained from randomized SVD for the three curvature feature extraction methods. The pseudo-Gaussian method exhibits pronounced information concentration: the first principal component is the largest, followed by an almost linear decline until the sixth component, after which the curve flattens, indicating that the major variations are captured by a small number of components. In contrast, the summation and mean methods show more dispersed information, with the first 15 components retaining relatively large eigenvalues and noticeable fluctuations, suggesting that the main information is not well concentrated. This demonstrates that the pseudo-Gaussian approach outperforms the other two in terms of feature concentration, dimensionality reduction efficiency, and potential discriminative power.

Furthermore, regardless of the method, the modes of variation are largely consistent between real and synthetic images. However, the differences between principal components of pseudo-Gaussian features for real versus synthetic images are significantly larger than those of the other methods.

A closer examination of the pseudo-Gaussian method reveals that real images generally have higher eigenvalues than synthetic ones, indicating that their curvature features exhibit greater variability and richer structure in the principal component space. Synthetic images, on the other hand, show lower eigenvalues, suggesting more uniform or smoother curvature patterns. This difference serves as a potential discriminative indicator for distinguishing real and synthetic images, and corroborates our earlier observation that synthetic images exhibit stronger velocity field consistency and more concentrated distributions.

We also note that, for the summation and mean methods, the eigenvalue curves of synthetic images are slightly higher. We attribute this to the fact that summation/mean operations tend to smooth out

local details, which suppresses extreme values in real images while amplifying the more uniform overall curvature of synthetic images, resulting in slightly higher values in the principal component space. In contrast, the pseudo-Gaussian method better preserves local curvature information, which we believe underlies the artifact-focused capability demonstrated in Section. 6.1.

### C.3 FORMAL THEORY OF CURVATURE-BASED DISCRIMINATION

In this section we formalize the role of curvature as a discriminative statistic for real vs. synthetic images in our PF–ODE framework. We proceed in two steps. (1) We show that the pseudo-Gaussian curvature statistic $\kappa_{\text{pseudo}}$ is a bounded continuous functional of the PF–ODE trajectory, so it is a well-posed and stable quantity. (2) We give a general guarantee that whenever a scalar statistic computed from $\kappa_{\text{pseudo}}$ has a population-level gap between real and synthetic trajectories, then the curvature distributions themselves must differ, and a curvature threshold classifier is strictly better than random guessing. We then explain why the pseudo-Gaussian aggregation is a natural choice among curvature-based statistics in our setting.

Define the trajectory:
$$\xi_{x_0}(t) = x_t \tag{42}$$

and recall from Sec. 4.2 the reparameterized trajectories $\bar{x}_t$ (Eq. 11), the finite-difference curvature surrogate $\tilde{\kappa}_t$ (Eq. 15), and the pseudo-Gaussian curvature $\kappa_{\text{pseudo}}$ (Eq. 16). Let $\text{C}([0,T]; \mathbb{R}^D)$ be the space of continuous curves endowed with the uniform norm. Let $\Phi$ denote the composition of the curvature surrogate and pseudo-Gaussian aggregation, so that

$$\kappa_{\text{pseudo}}(x_0) = \Phi(\xi_{x_0}) \tag{43}$$

### C.3.1 CURVATURE AS A CONTINUOUS FUNCTIONAL OF PF-ODE TRAJECTORIES

This section makes precise that $\kappa_{\text{pseudo}}$ is a continuous and bounded functional of the PF-ODE trajectory. Intuitively, small perturbations of the initial condition move the whole trajectory slightly (in sup-norm), and the curvature construction uses only finite differences and continuous algebraic operations. Therefore, $\kappa_{\text{pseudo}}$ behaves stably.

We first state a structural property that collects the regularity assumptions implicitly used in Sec. 4.1 and Sec. 4.2.

**Assumption 1.** *The velocity field $v_\theta(x,t)$ is locally Lipschitz in $x$ and continuous in $t$, uniformly on compact sets. The time reparameterization $\gamma_t$ is continuously differentiable and strictly monotone. The score network $\epsilon_\theta(x,t)$ is continuous and bounded on compact sets, and its norm is bounded away from zero on the compact trajectory set induced by $x_0 \sim p_0, q_0$.*

**Proposition 1.** *Under Assumption 1, the map $x_0 \mapsto \kappa_{\text{pseudo}}(x_0) = \Phi(\xi_{x_0})$ is continuous in the initial condition $x_0$. Moreover, $\kappa_{\text{pseudo}}$ is bounded on any compact domain in $\mathbb{R}^D$.*

*Proof. Step 1: Continuous dependence of PF-ODE trajectories on the initial condition.*

By Assumption 1, the PF-ODE vector field is locally Lipschitz in $x$. Hence, for every initial condition $x_0$ there exists a unique solution $\xi_{x_0} : [0,T] \to \mathbb{R}^D$ by the Picard–Lindelöf theorem. For any two initial conditions $x_0, y_0$, Grönwall's inequality gives the standard stability estimate

$$\|\xi_{x_0}(t) - \xi_{y_0}(t)\| \leqslant \|x_0 - y_0\| \exp(Lt), \quad t \in [0,T] \tag{44}$$

where $L$ is a Lipschitz constant on the relevant compact set. Taking the supremum over $t \in [0,T]$ yields

$$\|\xi_{x_0} - \xi_{y_0}\|_\infty := \sup_{t \in [0,T]} \|\xi_{x_0}(t) - \xi_{y_0}(t)\| \leqslant e^{LT} \|x_0 - y_0\| \tag{45}$$

Thus the solution map $x_0 \mapsto \xi_{x_0}$ is Lipschitz continuous from $\mathbb{R}^D$ into $\text{C}([0,T]; \mathbb{R}^D)$.

*Step 2: Continuity of the curvature surrogate in the trajectory.*

Fix a time discretization $0 = t_0 < t_1 < \cdots < t_T = T$. For each trajectory $\xi \in \text{C}([0,T]; \mathbb{R}^D)$, the discrete samples

$$\bar{x}_{t_k} := \xi(t_k), \quad k = 0, \ldots, T \tag{46}$$

are evaluations of a continuous linear functional. Finite difference operators of the form

$$D_\xi(t_k) := \frac{\xi(t_{k+1}) - \xi(t_k)}{\Delta t_k} \tag{47}$$

and higher order differences obtained by iterating this construction are continuous linear maps on $\mathrm{C}([0,T];\mathbb{R}^D)$. The curvature surrogate $\tilde{\kappa}_{t_k}$ is built from the discrete samples $\bar{x}_{t_k}$, the score evaluations $\epsilon_\theta(x_{t_k}, t_k)$ (continuous under Assumption 1), and algebraic operations such as inner products, norms, and rational expressions. All these operations are continuous. The lower bound on $\|\epsilon_\theta(x,t)\|$ in Assumption 1 guarantees that denominators in the rational expressions are bounded away from zero on the compact trajectory set, so no singularities appear. Hence, the map

$$\xi \mapsto (\tilde{\kappa}_{t_k})_{k=0}^T \tag{48}$$

is continuous from $\mathrm{C}([0,T];\mathbb{R}^D)$ to $\mathbb{R}^{T+1}$.

*Step 3: Continuity and boundedness of the pseudo-Gaussian functional.*

The pseudo-Gaussian aggregation

$$\Phi : \mathbb{R}^{T+1} \to \mathbb{R}, \qquad (\tilde{\kappa}_{t_k})_{k=0}^T \mapsto \left(\max_k \tilde{\kappa}_{t_k}\right) \cdot \left(\min_k \tilde{\kappa}_{t_k}\right) \tag{49}$$

is continuous on the finite-dimensional space $\mathbb{R}^{T+1}$, since it is composed of coordinatewise maxima and minima together with multiplication. Combining Step 1 and Step 2 with this observation, we obtain that

$$x_0 \mapsto \kappa_{\mathrm{pseudo}}(x_0) = \Phi(\xi_{x_0}) \tag{50}$$

is continuous as a composition of continuous maps. For boundedness, let $B \subset \mathbb{R}^D$ be compact. By Step 1, the set of trajectories $\{\xi_{x_0} : x_0 \in B\}$ is contained in a compact subset of $\mathrm{C}([0,T];\mathbb{R}^D)$. Under Assumption 1, the score and all coefficients entering the definition of $\tilde{\kappa}_{t_k}$ are bounded on this compact set. Since $\Phi \circ \xi(\cdot)$ is continuous on the compact set $B$, it attains its maximum and minimum on $B$. In particular, $\kappa_{\mathrm{pseudo}}$ is bounded on $B$. $\square$

According to the above proof, $\kappa_{\mathrm{pseudo}}$ is a stable, well-posed statistic of PF-ODE trajectories. This enables the distributional reasoning we develop next.

### C.3.2 CURVATURE DISTRIBUTIONS MUST DIFFER BETWEEN REAL AND SYNTHETIC IMAGES

We now connect trajectory-level differences with second-order curvature statistics. Let

$$\mathbb{P}_{real} := (\xi_\cdot)_\# p_0, \qquad \mathbb{P}_{syn} := (\xi_\cdot)_\# q_0 \tag{51}$$

be the induced trajectory laws on $\mathrm{C}([0,T];\mathbb{R}^D)$.

We consider bounded measurable trajectory statistics $S : \mathrm{C}([0,T];\mathbb{R}^D) \to \mathbb{R}$ such that

$$\mathbb{E}_{\mathbb{P}_{real}}[S] \neq \mathbb{E}_{\mathbb{P}_{syn}}[S] \tag{52}$$

For curvature-based statistics, we focus on the case where $S$ is deterministically obtained from the pseudo-Gaussian curvature.

**Assumption 2** (Curvature-based statistic). *There exists a bounded Borel function $h : \mathbb{R} \to \mathbb{R}$ such that*

$$S(\xi) = h\big(\kappa_{pseudo}(\xi)\big)$$

*where $\kappa_{pseudo}(\xi)$ is the pseudo-Gaussian curvature defined in Eq. 16.*

Assumption 2 simply states that the statistic of interest is computed from curvature, which is the case for our PF-ODE curvature detector.

**Assumption 3** (Non-degeneracy of curvature). *At least one of the real or synthetic curvature distributions is non-degenerate:*

$$\mathrm{Var}_{x_0 \sim p_0}\big[\kappa_{pseudo}(x_0)\big] > 0 \quad or \quad \mathrm{Var}_{x_0 \sim q_0}\big[\kappa_{pseudo}(x_0)\big] > 0$$

*Equivalently, $\kappa_{pseudo}(x_0)$ is not almost surely constant under both $p_0$ and $q_0$.*

**Proposition 2.** *Under Assumptions 1, 2 and 3, if Eq. 52 holds, then the push-forward curvature distributions satisfy*

$$(\kappa_{pseudo})_{\#}p_0 \neq (\kappa_{pseudo})_{\#}q_0$$

*Consequently, there exists a threshold $\tau \in \mathbb{R}$ such that*

$$\mathbb{P}_{p_0}\big(\kappa_{pseudo} \geq \tau\big) \neq \mathbb{P}_{q_0}\big(\kappa_{pseudo} \geq \tau\big)$$

*so a threshold classifier based on $\kappa_{pseudo}$ is strictly better than random guessing.*

*Proof.* Let

$$\mu_{real} := (\kappa_{\text{pseudo}})_{\#}\mathbb{P}_{real}, \qquad \mu_{syn} := (\kappa_{\text{pseudo}})_{\#}\mathbb{P}_{syn} \tag{53}$$

be the curvature push-forward measures on $\mathbb{R}$. By Assumption 2,

$$\mathbb{E}_{\mathbb{P}_{real}}[S] = \mathbb{E}_{\mathbb{P}_{real}}[h(\kappa_{\text{pseudo}})] = \int h(\kappa)\,\mu_{real}(d\kappa), \quad \mathbb{E}_{\mathbb{P}_{syn}}[S] = \int h(\kappa)\,\mu_{syn}(d\kappa) \tag{54}$$

Suppose for contradiction that $\mu_{real} = \mu_{syn}$. Then for every bounded Borel $h$ we must have

$$\int h(\kappa)\,\mu_{real}(d\kappa) = \int h(\kappa)\,\mu_{syn}(d\kappa) \tag{55}$$

and in particular for the specific $h$ in Assumption 2 we obtain

$$\mathbb{E}_{\mathbb{P}_{real}}[S] = \mathbb{E}_{\mathbb{P}_{syn}}[S] \tag{56}$$

contradicting Eq. 52. Thus $\mu_{real} \neq \mu_{syn}$, which is exactly

$$(\kappa_{\text{pseudo}})_{\#}p_0 \neq (\kappa_{\text{pseudo}})_{\#}q_0 \tag{57}$$

Since these are laws of real-valued random variables, $\mu_{real} \neq \mu_{syn}$ implies the existence of a threshold $\tau$ such that

$$\mathbb{P}_{p_0}(\kappa_{\text{pseudo}} \geq \tau) \neq \mathbb{P}_{q_0}(\kappa_{\text{pseudo}} \geq \tau) \tag{58}$$

which yields a classifier strictly better than random guessing. □

The proposition shows that whenever a population-level gap can be captured by a curvature-based statistic $S = h(\kappa_{\text{pseudo}})$, the induced curvature distributions for real and synthetic images cannot coincide.

# D  DATA

This section provides detailed information about the datasets, including both the training and test splits. The specific data used in our experiments will be released after the camera-ready version.

## D.1  TRAINING DATA

We train our method on SD + LAION. For the synthetic SD images, we randomly select 40k generated samples from the DiffusionDB (Wang et al., 2023b) dataset. For the real images, we randomly select 40k samples from LAION (Schuhmann et al., 2021) with predicted aesthetic scores of 6.25 or higher.

## D.2  EVALUATION DATA

This section describes the construction of our evaluation data and the parameters used for each generative model. For Imagen (Saharia et al., 2022), Midjourney (Midjourney, 2022), and DALL·E 3, the corresponding real images are retrieved using the RIS algorithm from FakeInversion (Cazenavette et al., 2024), which searches the Internet for real images published before January 1, 2021 that are visually and semantically similar to the synthetic ones.

For all other models, synthetic images are generated using COCO (Lin et al., 2014) prompts. Each model is assigned an independent set of prompts, and no COCO prompt is shared across models. The real images associated with each prompt in COCO form the real dataset corresponding to that model. This construction ensures that all datasets are mutually exclusive.

### D.2.1 KANDINSKY 2

For our Kandinsky 2 images, we use the Kandinsky 2.2 (Shakhmatov et al., 2023) model from Hugging Face (link), using the default parameters given in their usage example:

- `prior_guidance_scale=1.0`
- `height=768`
- `width=768`
- `negative_prompt="low quality, bad quality"`

### D.2.2 KANDINSKY 3

For our Kandinsky 3 images, we use the Kandinsky 3 (Arkhipin et al., 2023) model from Hugging Face (link), using the default parameters given in their usage example:

- `num_inference_steps=50`

### D.2.3 PIXART-$\alpha$

For our PixArt-$\alpha$ (Chen et al., 2023) images, we use the 1024 resolution model from Hugging Face (link). All parameters are left as their defaults.

### D.2.4 PLAYGROUND 2.5

For our Playground 2.5 images, we use the Playground 2.5 (Li et al., 2024) model from Hugging Face (link), using the default parameters given in their usage example:

- `num_inference_steps=50`
- `guidance_scale=3`

### D.2.5 SDXL DIRECT PREFERENCE OPTIMIZATION

For our SDXL-DPO images, we use the SDXL-DPO (Wallace et al., 2024) model from Hugging Face (link), with the default parameters given in their usage example:

- `guidance_scale=5`

### D.2.6 STABLE DIFFUSION XL

For our SDXL images, we use the Stable Diffusion XL (Podell et al., 2023) model from Hugging Face (link), using both the base and refiner models with the default parameters given in their usage example:

- `num_inference_steps=40`
- `denoising_end=0.8`
- `denoising_start=0.8`

### D.2.7 SEGMIND MIXTURE OF EXPERTS

For our Seg-MoE images, we use the SegMoE-4x2-v0 (Harish et al., 2024) model from Hugging Face (link), with the default parameters given in their usage example:

- `negative_prompt="nsfw, worse quality, bad quality"`
- `height=1024`
- `width=1024`
- `num_inference_steps=25`
- `guidance_scale=7.5`

### D.2.8 SEGMIND STABLE DIFFUSION 1B

For our SSD-1B (Gupta et al., 2024) images, we use the SSD-1B model from Hugging Face (link), using the default parameters given in their usage example:

- `negative_prompt="ugly, blurry, poor quality"`

### D.2.9 STABLE CASCADE

For our Stable Cascade (Pablo et al., 2023) images, we use the Stable Cascade model from Hugging Face (link), using the default parameters given in their usage example for the prior model:

- `height=1024`
- `width=1024`
- `guidance_scale=4.0`
- `num_inference_steps=20`

and decoder model:

- `guidance_scale=0.0`
- `num_inference_steps=10`

### D.2.10 SEGMIND VEGA

For our Segmind Vega (Gupta et al., 2024) images, we use the Segmind Vega model from Hugging Face (link), using the default parameters given in their usage example:

- `negative_prompt="(worst quality, low quality, illustration, 3d, 2d, painting, cartoons, sketch)"`

### D.2.11 WÜRSTCHEN 2

For our Würstchen (Pablo et al., 2023) images, we use the Würstchen v2 model from Hugging Face (link), using the default parameters given in their usage example:

- `height=1024`
- `width=1024`
- `prior_guidance_scale=4.0`
- `decoder_guidance_scale=0.0`

### D.2.12 FLUX

For our FLUX (Pablo et al., 2023) images, we use the FLUX model from Hugging Face (link), using the default parameters given in their usage example:

- `guidance_scale=0.0`
- `num_inference_steps=4`
- `max_sequence_length=256`

### D.2.13 STABLE DIFFUSION 3

For our SD3 (Esser et al., 2024) images, we use the Stable Diffusion v3 model from Hugging Face (link), using the default parameters given in their usage example:

- `negative_prompt="ugly, blurry, poor quality"`
- `num_inference_steps=28`
- `guidance_scale=7.0`

| Eval Set | CNNDet | | DMDet | |
|---|---|---|---|---|
| | ACC | AUCROC | ACC | AUCROC |
| DALL·E 2 | 0.624 | 0.680 | 0.618 | 0.672 |
| DALL·E 3 | 0.659 | 0.716 | 0.461 | 0.415 |
| Midjourney v5/6 | 0.595 | 0.630 | 0.485 | 0.484 |
| Imagen | 0.674 | 0.714 | 0.521 | 0.573 |
| Kandinsky 2 | 0.574 | 0.600 | 0.483 | 0.478 |
| Kandinsky 3 | 0.609 | 0.659 | 0.588 | 0.614 |
| PixArt-$\alpha$ | 0.591 | 0.627 | 0.523 | 0.580 |
| Playground 2.5 | 0.553 | 0.582 | 0.502 | 0.517 |
| SDXL-DPO | 0.761 | 0.843 | 0.515 | 0.563 |
| SDXL | 0.735 | 0.814 | 0.549 | 0.568 |
| Seg-MOE | 0.625 | 0.663 | 0.480 | 0.476 |
| SSD-1B | 0.665 | 0.726 | 0.583 | 0.556 |
| Stable-Cascade | 0.652 | 0.705 | 0.539 | 0.565 |
| Segmind Vega | 0.676 | 0.742 | 0.564 | 0.540 |
| Würstchen 2 | 0.580 | 0.610 | 0.640 | 0.675 |
| ADM | 0.582 | 0.740 | 0.697 | 0.746 |
| Glide | 0.580 | 0.732 | 0.784 | 0.857 |
| VQDM | 0.552 | 0.671 | 0.528 | 0.520 |
| FLUX | 0.498 | 0.540 | 0.512 | 0.603 |
| Stable Diffusion 1.4 | 0.502 | 0.558 | 0.599 | 0.702 |
| Stable Diffusion 1.5 | 0.512 | 0.603 | 0.585 | 0.653 |
| Stable Diffusion 3 | 0.506 | 0.570 | 0.590 | 0.644 |
| **Average** | 0.605 | 0.669 | 0.561 | 0.591 |

Table 5: Results (**ACC / AUCROC**) of CNNDet (Wang et al., 2020) and DMDet (Corvi et al., 2023b) evaluated on our collected diffusion models and updated generative models. All the modles are re-trained with the official codes. This is a supplement to Table 1 in the main text.

# E IMPLEMENTATION DETAILS

Our experiments are conducted on an A800 GPU for a total of 15 epochs. All images are center-cropped to 64×64 before being fed into the model. Random horizontal flipping and rotation are applied as data augmentation. We use the AdamW optimizer, setting the learning rate of the second-order feature pipeline to 1e-5 with a weight decay of 0.05, the zeroth-order feature pipeline to 1e-4 with a weight decay of 0.01, and the shared projection layer and classifier to 1e-3 with zero weight decay. The learning rates are decoupled to account for the differing sensitivities of features at different orders to input variations.

It is worth noting that the selected ODE pipeline operates in the RGB domain, and all conclusions in this work are based on RGB images. However, we anticipate that our approach could be extended to ODE pipelines in latent spaces, potentially supporting larger receptive fields, which remains an avenue for future research.

# F SUPPLEMENT TO THE MAIN EXPERIMENT

Due to space limitations, we report the detection results of CNNDet (Wang et al., 2020) and DMDet (Corvi et al., 2023b) across all datasets in this section. Accordingly, Table 5 serves as a supplement to Table 1 in the main paper.

