# OpenReview forum: "Synthetic Image Detection via Curvature of Diffusion Probability Flows"
_ICLR.cc/2026/Conference — Submitted to ICLR 2026_

### Official Review · Reviewer_SN55 · 2025-10-25

**Soundness:** 2
**Presentation:** 2
**Contribution:** 2
**Rating:** 4
**Confidence:** 4

**Summary:**

The paper tackles the task of synthetic image detection. Specifically, the authors investigate the effect of curvature information from diffusion model ODE trajectories for this purpose. The proposed method is evaluated with several competitors and the authors claim SOTA results with a more efficient approach compared to diffusion inversion.

**Strengths:**

1. Overall, the pipeline is clearly described and the methodology is clear.
2. The attempt at a more rigorous and precise treatment of the ODE trajectories and curvature is appreciated.
3. The authors show improved performance on standard benchmarks. TPRs @ 5% FPR are also included for a more thorough analysis.
4. The authors provide ablations on NFEs and training sets.

**Weaknesses:**

1. I found the paper vague or unclear on various aspects. Please see the Questions section below for specific concerns.

2. The Section on interpretability (line 458) is poorly explained and not particularly convincing. The authors mention "In synthetic
images, areas with inconsistent lighting, incorrect
perspective, or structural anomalies receive the highest saliency". However, I struggle to see any examples of this in Figure 5.

**Questions:**

1. The abstract reads "We show that the discrete-step distances on the Wasserstein manifold inherently encode reconstruction error" and discussion in Section 4.1 mentions that existing observations regarding the overall reconstruction error, when combined with Theorem 2, imply that synthetic trajectories tend to be straighter than those of real images (line 236). As I understand, Theorem 2 links W2 between intermediate marginals to the velocity, saying nothing about the reconstruction error obtained via inversion. Can the authors clarify?

2. As I understand, curvature features are extracted over the context of the full trajectories (Equation 16). Some discussion on this choice and whether your findings connect with existing work (e.g., [1]) would be appreciated. Have you investigated whether specific regions/time-steps are consistently more informative?

3. The overall method includes curvature features and wavelet features. Does the method not work without them? An ablation study on this, e.g., curvature only or wavelet only or both would be helpful.

4. How should the results in Tables 1 and 2 be interpreted? As I understand, you report conditional metrics on synthetic data from various models. This not sufficient to judge performance without also quantifying performance on real data. While you have provided details about the synthetic data, I couldn't find any information about benchmarking on real data, which is just as important in binary classification. Could you please clarify?


[Minor] Consider merging the Appendix (in supplementary material) with the main paper for better readability.

[Minor] The paper is mathematically dense so consider being a bit more explicit with notation and definitions. For example, it would be helpful to include the definition of geometric curvature and the derivation for the particular ODE resulting in Equation 12 in the Appendix.

[1] Tracing the Roots: Leveraging Temporal Dynamics in Diffusion Trajectories for Origin Attribution, 2024

---

> ### Author Response · Authors · 2025-11-26
> **Official Comment by Authors**
>
> ### **W1:  The paper vague or unclear on various aspects.**
> Thank you for this feedback. In the `revised version`, we have clarified the **data construction**, **the main theoretical statements**, and the **datasets protocol** in response to the points raised in the Questions section, so that the presentation is more precise and easier to follow.
>
> ---
> ### **W2:  The Section on interpretability (line 458) is poorly explained and not particularly convincing.**
> Thank you for this comment. We agree that our original wording was stronger than what Figure 5 can visually support. Modern generative models produce very high-quality images, so the artifacts our detector focuses on are often subtle and not always obvious to the human eye. In the revised version, we have softened the interpretability claim: **we now state only that the saliency in synthetic images tends to concentrate on localized structural details, suggesting that the model relies on fine-grained, region-specific cues rather than purely global statistics.** We have also renamed the subsection from **“Interpretability” to `Saliency Analysis`** to better reflect that this part is a qualitative saliency-based visualization rather than a full interpretability study.
>
> ---
> ### **Q1:  Clarification on the link between Theorem 2 and reconstruction error**
> We thank the reviewer for raising this point. We agree that `Theorem 2`, in its mathematical form, characterizes an upper bound on the cumulative W₂ displacement along the ODE trajectory and does not directly define the reconstruction error obtained by explicit diffusion inversion.
>
> **Our intention is not to claim equivalence, but rather the following two-step implication, which we now clarify:**
> 1. **`Theorem 2` states that the cumulative discrete W₂ distances are upper-bounded by the total kinetic energy of the velocity field.** Since the PF-ODE velocity is fully determined by the score network, this quantity is directly computable from the model.
>
> 2. **The non-optimality term introduced in `Eq. (10)` bridges this bound to reconstruction error.** `Eq. (10)` measures the gap between the kinetic-energy upper bound (from `Theorem 2`) and the actual one-step Wasserstein displacement. Summing these gaps yields a **cumulative non-optimality $\sum E_t$​, which serves as a surrogate measure of how far the trajectory deviates from the optimal transport path** preferred by the model.
>
> Empirically (`Fig. 3`), real images accumulate larger non optimality than synthetic ones. Consequently, during the PF ODE mapping, **real samples** undergo a trajectory pattern **$p_0 \\rightarrow \\mathcal{N}(0, I) \\rightarrow q_0$**, while **synthetic samples** follow **$q_0 \\rightarrow \\mathcal{N}(0, I) \\rightarrow q_0$**. This indicates that real images deviate more strongly from the model’s low energy transport geometry. **This deviation is what reconstruction based detectors are sensitive to and manifests as reconstruction error.**
>
> Therefore, our claim is that **the discrete W₂ distances and their kinetic energy bound (`Theorem 2`), when combined with the computable non optimality term in `Eq. (10)`, inherently encode a trajectory deviation effect that is closely related to reconstruction error**, without requiring full inversion. We hope this clarifies the connection. Meanwhile, we have also updated the `abstract` and the relevant sentences in the `main text` to avoid any misleading impression of a strict equivalence between discrete Wasserstein distances and reconstruction error.

---

> ### Author Response · Authors · 2025-11-26
> **Official Comment by Authors**
>
> ### **Q2: Regarding whether specific regions/time steps are more informative and about the relation to [1].**
> **Whether specific regions/time steps:**
>
> We have indeed investigated this question during development. Motivated by the non-optimality term Et in `Eq. (10)`, we first analyzed how Et accumulates over time. Empirically, for both real and synthetic images **we observe that Et tends to be largest around the middle of the normalized PF-ODE time interval, while it is noticeably smaller near the beginning and the end.** This suggests, at first sight, that restricting curvature features to a central time window could be beneficial.
>
> However, in practice we cannot jump directly to the middle of a PF-ODE trajectory: **all intermediate marginals must still be obtained by integrating from x0.** Every additional ODE step increases the computational cost, and discarding curvature features computed from earlier marginals does not reduce the number of function evaluations. Since our pipeline models an instantaneous velocity field rather than a time-averaged field, capturing curvature reliably is also simpler and more stable when we use a globally uniform step size over the chosen time interval. Moreover, although the average Et is larger around the middle, we still observe occasional sharp curvature spikes away from this region, so aggressively truncating to a narrow central window risks losing informative events.
>
> We have also experimented with non-uniform temporal sampling schemes, including (i) using shorter fixed windows around the middle portion of the trajectory, and (ii) sampling time points from Gaussian distributions over [0,1] centered at t=0.5 with different variances, while keeping the total number of sampled time points fixed. None of these variants produced consistent improvements over the uniform, equal-step sampling strategy reported in the paper, while they offered no computational advantage because the PF-ODE still had to be integrated from the initial time. **We therefore chose to aggregate curvature over a uniformly sampled trajectory segment, which we found to be the best trade-off between discriminative performance and efficiency.**
>
> **About the relation to [1]:**
>
> We would like to clarify that our method was not derived from the temporal-dynamics framework in [1]. The approach in this paper was originally motivated by the line of reconstruction–error based detectors (e.g., DIRE, AEROBLADE, FakeInversion, LaRE2, FIRE), and our main question was whether the geometric properties of the PF-ODE itself already encode the discrepancy that these methods measure via explicit inversion. This line of reasoning led us to study kinetic energy, non-optimality (`Eq. (10)`), and finally curvature features of the probability flow.
>
> But the above empirical finding is also consistent with [1], which similarly reports that using global temporal context along the diffusion trajectory is more effective than relying on a single “golden” time step or a narrow temporal window.
>
> **The connection to [1] is therefore conceptual and complementary rather than inspirational:** we see [1] as independent evidence that temporal diffusion trajectories carry rich discriminative information, while our work explores a different geometric statistic (curvature in Wasserstein space). In the `revised version`, we have added a short discussion of this connection, together with the corresponding citation to [1], in the `Sec.4.2`.
>
> > [1] Tracing the Roots: Leveraging Temporal Dynamics in Diffusion Trajectories for Origin Attribution, 2024

---

> ### Author Response · Authors · 2025-11-26
> **Official Comment by Authors**
>
> ### **Q3: Curvature vs. wavelet features**
> Thank you for raising this point. A similar question was also raised by another reviewer. We have **added an explicit ablation in `Table 3`** of the `revised version` that separates the contributions of curvature and wavelet features.
>
> Concretely, using **only the curvature** branch yields **0.803 / 0.928 (ACC / AUCROC)**, while using **only the wavelet branch with the HH subband** yields **0.796 / 0.902**. Both variants are clearly weaker than the **full model**, which reaches **0.939 / 0.981** when the two branches are fused. This shows that the method does work with curvature-only or wavelet-only features, but each branch alone is substantially less accurate than the combined model.
>
> **We also clarify this interpretation in `Sec. 6.1 (Ablation of Curvature and Wavelet)`.** Curvature features measure the temporal smoothness of probability flow trajectories. They are sensitive to how easily a sample is transported toward the Gaussian prior. In contrast, HH wavelet features emphasize diagonal edges and fine textures in image space, and they respond strongly to local synthesis artifacts. The large gain from combining the two (from ≈0.80 to 0.94 ACC) indicates that they capture complementary rather than redundant information, and that both components are useful for the overall performance.
>
> | Only | ACC / AUCROC |
> |----------|--------------|
> | Cur. | 0.803 / 0.928 |
> | DWT (Yl) | 0.637 / 0.687 |
> | DWT (LH) | 0.707 / 0.803 |
> | DWT (HL) | 0.779 / 0.864 |
> | DWT (HH) | 0.796 / 0.902 |
>
> **Table 3.** Ablation results ACC / AUCROC on curvature only or DWT only.
>
> ---
> ### **Q4: The information of data construction, especially for real data.**
> Thank you for pointing this out. All metrics in `Table 1` and `Table 2` (ACC, AUCROC, and TPR@5%FPR) are computed on the full binary task between real and synthetic images. For each generator, **we construct a paired real and fake dataset**, and ACC and AUCROC are measured over both classes. Thus, the reported numbers already reflect performance on real data as well as on synthetic data.
>
> **To avoid confusion, we now describe the construction of the datasets more explicitly in `Appendix D (Data)`.** Training uses only SD plus LAION. Concretely, we sample 40k synthetic images from DiffusionDB and 40k real images from LAION, filtered by a predicted aesthetic score threshold. For evaluation, all datasets are built independently of this training set. For Imagen, Midjourney, and DALL·E 3, the corresponding real images are retrieved using the RIS algorithm from FakeInversion, which searches for visually and semantically similar images that were published before 1 January 2021. For all other models, synthetic images are generated from COCO prompts. Each model is assigned its own subset of COCO prompts, and the COCO images associated with these prompts form the corresponding real sets.
>
> ---
> ### **[Minor1] On merging the appendix with the main paper.**
> Thank you for pointing this out. You are right that the ICLR guidelines recommend submitting a single file containing the paper and the supplementary text, with the appendix placed after the references. We now follow the official instructions and provide a single unified PDF.
>
> ---
> ### **[Minor2] On notation and definitions / Eq. (12).**
> Thank you for this helpful suggestion. In the revised version, we make the mathematical exposition more explicit. First, we now provide the **fundamental definition of geometric curvature** together with the **detailed derivation** of the PF-ODE–specific form leading to `Eq. (12)` in `Appendix C.1`. Second, around several dense equations we **add short verbal explanations** to clarify their meaning. For example, right after `Eq. (10)` we now explicitly interpret the term $E_t$ as a stepwise non-optimality measure and its cumulative sum as a surrogate measure of how far the trajectory deviates from the optimal transport path. We hope these additions make the notation and derivations easier to follow.

---

### Official Review · Reviewer_UghK · 2025-10-28

**Soundness:** 3
**Presentation:** 3
**Contribution:** 3
**Rating:** 6
**Confidence:** 3

**Summary:**

This paper introduces a novel approach for Synthetic Image Detection (SID) that leverages the curvature characteristics of diffusion probability flow ODE trajectories rather than relying on full reconstruction. The idea is that real and synthetic images follow distinct diffusion trajectories — real images exhibit higher curvature variance and higher-energy paths, while synthetic ones produce smoother, lower-energy trajectories during diffusion inversion. Building on the curvature analysis, the paper propose a classification pipeline for SID that leverages features based on a pseudo-Gaussian curvature descriptor and  wavelet features to capture fine-grained spatial cues. Experiments demonstrate strong performance over prior state-of-the-art methods, such as B-Free and FakeInversion.

**Strengths:**

1. Novel formulation of SID via optimal transport and probability flow curvature, linking reconstruction error to Wasserstein geometry and ODE kinetic energy.

2. Theorems 1–2 coherently connect Wasserstein bounds with velocity field energy, justifying curvature-based descriptors.

3. Convincing performance gains on the final benchmark

**Weaknesses:**

1. Potential overfitting or data leakage: The claim of generalization from training to unseen models is strong, but the text doesn’t detail whether prompt overlap or data source contamination might occur between training and evaluation sets and also the opensource model used for obtaining synthetic training data.

2. The jump from Eq. 10 (non-optimality term) to the claim that "synthetic images lie on manifold regions more easily represented by the model" lacks rigorous justification.

3. Table 2 shows that performance drops significantly with 5 NFEs and plateaus beyond 10, but there's no explanation of why or how this relates to the theoretical framework.

4. No comparative studies on the contribution to the performance from the curvature features vs that from the wavelet features.

5. Although the paper claims “less than half the computational cost,” no empirical nor theoretical runtime or FLOPs comparison or estimation is provided.

**Questions:**

1. Choice of curvature: why choose the pseudo-Gaussian curvature? Fig 2(c) - mean of \tilde{k}_t - seems to also be a discriminative alternative?

2. Can you provide ablation studies separating the contribution of curvature features vs. wavelet features?

**Details Of Ethics Concerns:**

No concerns.

---

> ### Author Response · Authors · 2025-11-26
> **Official Comment by Authors**
>
> ### **W1:  Potential overfitting or data leakage.**
> Thank you for raising this point. We agree that generalization claims require a clear separation between training and evaluation data. In the revised version **we explicitly describe our data construction protocol in `Appendix D (Data)`.** We construct training and evaluation data from disjoint sources.
>
> **For training**, we use **SD plus LAION** only. Concretely, we sample 40k synthetic images from DiffusionDB[1] and 40k real images from LAION[2] with predicted aesthetic scores of at least 6.25. These images are used exclusively for training and do not overlap with any evaluation set.
>
> **For evaluation**, we construct all datasets separately from the training data. For Imagen, Midjourney, and DALL·E 3, the corresponding real images are retrieved using the RIS algorithm from FakeInversion[3], which searches for visually and semantically similar images **published before 1 January 2021**. For all other models, synthetic images are generated from COCO prompts. Each model is assigned an independent subset of COCO prompts, with **no prompt shared across models**, and the COCO images associated with these prompts form the corresponding real sets.
>
> Finally, we note that we cannot fully trace the proprietary or web scale training data used to train Stable Diffusion or the other generative models. Our goal in this work is to assess cross model generalization under realistic conditions, instead of trying to perfectly decontaminate the underlying large scale internet training data. Any incidental overlap at that level is unavoidable in practice and makes the evaluation closer to real world deployment scenarios.
>
> > [1] Diffusiondb: A large-scale prompt gallery dataset for text-to-image generative models. ACL. 2023.
>
> > [2] Laion-400m: Open dataset of clip-filtered 400 million image-text pairs. 2021.
>
> > [3] Fakeinversion: Learning to detect images from unseen text-to-image models by inverting stable diffusion. CVPR. 2024.

---

> ### Author Response · Authors · 2025-11-26
> **Official Comment by Authors**
>
> ### **W2:  The jump from Eq. 10 (non-optimality term) to the claim that "synthetic images lie on manifold regions more easily represented by the model" lacks rigorous justification.**
> Thank you for pointing this out. Our intention was not to claim that `Eq. 10` alone mathematically proves that synthetic images lie in an “easier” manifold region. Rather, `Eq. 10` is used to define a non-optimality measure that we interpret in the light of optimal transport and Wasserstein gradient flows.
>
> **What `Eq. 10` measures.**
> `Eq. 10` defines the stepwise non optimality term, which is the gap between the kinetic energy based upper bound from Theorem 1 and the actual one step $W_2$ distance. Summing over t yields $\sum E_t$, which indicates whether the path behaves more like a low-energy transport path. Trajectories with small cumulative $\sum E_t$ are therefore closer to the optimal transport path in Wasserstein space.
>
> **Connection to “easier to represent” regions.**
> During training, the diffusion model is optimized so that its velocity field approximates the Wasserstein gradient flow of the KL functional with respect to the data distribution. Consider distributions that are well aligned with the model, meaning that they lie in regions of high model likelihood. For such distributions, the induced probability flow trajectories are expected to stay closer to the minimal-energy gradient flow paths. As a result, they exhibit smaller cumulative non-optimality. Conversely, distributions that are harder for the model to represent tend to produce trajectories with larger deviations and hence larger $\sum E_t$.
>
> **Empirical evidence.**
> In practice, `Figure 2` and `Figure 3` show that synthetic images have both lower total kinetic energy and smaller cumulative non optimality than real images over the first half of the diffusion inversion, while mapping to similar Gaussian endpoints. This observation is consistent with prior work[3][4][5][6]. These studies report that synthetic images achieve lower reconstruction errors than real images under the same reconstruction pipeline. Taken together, these results support the view that synthetic samples tend to follow lower energy and more geodesic like trajectories under the learned PF-ODE. Thus, `Eq. (10) ` itself defines a general non-optimality measure, and in our experiments we observe that this measure consistently separates real and synthetic trajectories, which is in line with prior findings on reconstruction errors.
>
> **Textual clarification in the revision.**
> To avoid overstating the theoretical claim, we have revised the manuscript to present this as an empirically grounded interpretation rather than a formal theorem. Specifically, the original sentence
>
> *“As a result, synthetic images consistently achieve smaller reconstruction errors, suggesting that they lie on manifold regions more easily represented by the model.”*
>
> **has been replaced by**
>
> *“In particular, synthetic images consistently achieve smaller cumulative non optimality on our benchmarks, which empirically suggests that their trajectories tend to remain closer to the low energy transport paths preferred by the model.”*
>
> **We also add a short explanation after `Eq. 10` clarifying that the cumulative non optimality $\sum E_t$ measures deviation from the optimal transport path, and we now phrase our interpretation in terms of low energy transport paths in Wasserstein space rather than manifold level “easier to represent” regions.**
>
> In addition, **we have updated the `abstract` and `introduction`** to use the same low energy transport path interpretation, so that the presentation is consistent across the entire manuscript. We hope this clarification addresses the concern and makes the connection between `Eq. 10` and our interpretation more transparent.
>
> > [3] Fakeinversion: Learning to detect images from unseen text-to-image models by inverting stable diffusion. CVPR. 2024.
>
> > [4] Dire for diffusion-generated image detection. CVPR. 2023.
>
> > [5] Aeroblade: Training-free detection of latent diffusion images using autoencoder reconstruction error. CVPR. 2024.
>
> > [6] Fire: Robust detection of diffusion-generated images via frequency-guided reconstruction error. CVPR. 2025.

---

> ### Author Response · Authors · 2025-11-26
> **Official Comment by Authors**
>
> ### **W3: Table 2 shows that performance drops significantly with 5 NFEs and plateaus beyond 10, but there's no explanation of why or how this relates to the theoretical framework.**
> Thank you for this comment. The dependence on the number of function evaluations (NFEs) is closely connected to how we approximate the continuous probability flow ODE and the geometric quantities used in our method.
>
> Our theoretical analysis is formulated in continuous time dynamic models, where the probability flow ODE induces a transport path on the Wasserstein manifold and both the non-optimality term and curvature are defined along this continuous trajectory. In practice we only have access to a discrete approximation of this path, obtained by integrating the PF ODE with a fixed step schedule. When the number of NFEs is small, for example **5 steps**, each step covers a large time interval. This leads to significant **local integration error** and a **coarse approximation** of the trajectory. As a result, both the discrete step distances that enter the non-optimality measure and the finite difference curvature surrogate become noisy and lose much of the geometric information predicted by the theory, which explains the noticeable performance drop at 5 NFEs.
>
> Conversely, when we increase NFEs **beyond 10**, the step size becomes small. In this regime, the discrete integration closely tracks the underlying continuous PF-ODE trajectory. Further increasing the number of steps only causes small numerical changes in the discrete step distances and in the finite-difference curvature estimates. These changes do not translate into noticeable gains in detection accuracy, which explains the plateau in `Table 2`. This behavior is consistent with the theoretical view: **once the discrete path already approximates the continuous low energy transport trajectory well, additional function evaluations mostly duplicate information while only increasing computational cost.**
>
> **We have clarified this point in the revised version by rewriting the corresponding sentence in `Sec.6.1 (Ablation of NFEs and Training set)`.**
>
> ---
> ### **W4&Q2: Ablation studies of curvature features vs. wavelet features.**
> Thank you for this comment. **In the `revised version` we add an explicit ablation in `Table 3`** that separates the contributions of curvature and wavelet features. Using **only the curvature** branch yields **80.3% ACC**, while using only the wavelet branch with the **HH subband** gives **79.6% ACC**. Both are much weaker than the **full model**, which reaches **93.9% ACC** when the two branches are fused. This shows that the two branches contribute comparable standalone performance but their combination brings a large additional gain, indicating that they capture complementary rather than redundant information.
>
> We also clarify this interpretation in the `Sec.6.1 (Ablation of Curvature and Wavelet)`. Curvature features measure the temporal smoothness of probability flow trajectories. They are sensitive to how easily a sample is transported toward the Gaussian prior. In contrast, HH wavelet features emphasize diagonal edges and fine textures in image space, and they respond strongly to local synthesis artifacts. Combining these global dynamical cues with local high frequency artifacts substantially improves accuracy, as reflected in `Table 3`.
>
> | Only | ACC / AUCROC |
> |----------|--------------|
> | Cur. | 0.803 / 0.928 |
> | DWT (Yl) | 0.637 / 0.687 |
> | DWT (LH) | 0.707 / 0.803 |
> | DWT (HL) | 0.779 / 0.864 |
> | DWT (HH) | 0.796 / 0.902 |
>
> **Table 3.** Ablation results ACC / AUCROC on curvature only or DWT only.

---

> > ### Author Response · Authors · 2025-11-26
> > **Official Comment by Authors**
> >
> > ### **W5:  Although the paper claims “less than half the computational cost,” no empirical nor theoretical runtime or FLOPs comparison or estimation is provided.**
> > Thank you for highlighting this issue. In the original manuscript, the phrase *“less than half the computational cost of full diffusion inversion”* was only stated qualitatively and was not backed up by explicit numbers. In the `revision`, we make this statement quantitative in terms of **FLOPs and parameter counts**. `Table 4` now reports these quantities for our method and for several strong reconstruction based baselines.
> >
> > Concretely, `Table 4` shows that full diffusion inversion detectors such as FakeInversion, LaRE$^2$ and FIRE require approximately **$1.0\times10^4$ B, $4.1\times10^4$ B and $3.3\times10^3$ B FLOPs** per image, respectively, whereas **our PF ODE based detector** needs only **$518.4$ B FLOPs**. We therefore restate the claim explicitly in terms of FLOPs. Our method uses less than half the FLOPs of diffusion process based full inversion approaches, while remaining comparable in FLOPs to the autoencoder-only method AEROBLADE and the non-reconstruction method B-Free. We have updated the `abstract` and `introduction` accordingly and now explicitly refer to `Table 4` as the quantitative justification for this efficiency claim.
> >
> > | Method        | Ref.       | #Params  | #FLOPs     |
> > |--------------|------------|----------|------------|
> > | **Non-reconstruction** | | | |
> > | UFD          | CVPR 2023  | 427.6 M  | 77.8 B     |
> > | NPR          | CVPR 2024  | 1.4 M    | 2.3 B      |
> > | FatFormer    | CVPR 2024  | 577.3 M  | 128.0 B    |
> > | B-Free       | CVPR 2025  | 85.5 M   | 553.1 B    |
> > | **Reconstruction-based** | | | |
> > | FakeInv.     | CVPR 2024  | 5.2 B    | 10385.0 B  |
> > | LaRE$^2$ | CVPR 2024 | 479.5 M | 41405.3 B |
> > | AEROBLADE    | CVPR 2024  | 84.0 M   | 446.0 B    |
> > | FIRE         | CVPR 2025  | 484.0 M  | 3346.2 B   |
> > | **Ours**     | -          | 298.4 M  | 518.4 B    |
> >
> > **Table 4.** We compare the number of model **parameters and FLOPs**. The upper part are non–reconstruction approaches, the lower part are reconstruction–based and the last one is our PF-ODE based method.
> >
> > ---
> > ### **Q1:  Choice of curvature: why choose the pseudo-Gaussian curvature? Fig 2(c) - mean of $\tilde{k}_t$ - seems to also be a discriminative alternative?**
> > Thank you for this question. Intuitively, **mean or sum** aggregation tends to **smooth out local curvature spikes** along the trajectory and is dominated by the average level of curvature. In contrast, our **pseudo-Gaussian curvature** uses the product of the maximal and minimal trajectory curvatures. This construction **emphasizes extreme bending events**, which are particularly informative for distinguishing real trajectories with irregular kinks from synthetic trajectories that are smoother.
> >
> > Although the mean based feature in `Fig. 2(c)` is somewhat discriminative, the histograms of real and synthetic images still show substantial overlap along the value axis. Curvature values in these overlapping regions are difficult to separate, and neither the L1 nor the L2 norm of the mean-based statistic can separate them effectively. Meanwhile, the PCA analysis in `Appendix C.2` shows that pseudo Gaussian features concentrate most of the discriminative variance in a very small number of principal components, and the resulting low dimensional projections more clearly separate real and synthetic samples than the mean based alternative.

---

### Official Review · Reviewer_KLwx · 2025-11-04

**Soundness:** 2
**Presentation:** 3
**Contribution:** 2
**Rating:** 4
**Confidence:** 3

**Summary:**

This paper proposes a reconstruction-free detector for synthetic image detection (SID) that works in the probability-flow ODE view of diffusion models. Instead of inverting a generator, the method computes finite-difference curvature features along a short **noising** trajectory and fuses them with diagonal high-frequency wavelet components. The authors claim strong cross-model generalization to **unseen** generators and **lower compute than full inversion**, reporting average ACC/AUCROC of **0.939 / 0.981** and **TPR@5%FPR = 0.933** on a broad suite of models.

**Strengths:**

1. **Good performance across many generators with an “unseen model” protocol.** The main table trains all detectors on **SD+LAION** and evaluates on other generators, showing strong averages and near-perfect numbers on several diffusion families. This explicitly targets cross-model generalization.
2. **Comprehensive core experiments and useful ablations.** The paper provides a broad comparison table and a steps/dataset ablation. The ablation indicates that **10–15 ODE steps** are a good operating point, with degradation at **5 steps** and diminishing returns at higher counts.
3. **Clear intuition that is practically helpful.** The authors motivate curvature via an OT/Wasserstein view: they argue that reconstruction-error signals are encoded by probability-flow energy and that *real* vs. *synthetic* images diverge most in the earlier half of inversion; their features focus on this regime. The intuition is coherent and guides the design, even if it is not fully proved to be discriminative (see Weaknesses).

**Weaknesses:**

1. **Lack of rigorous theory that curvature itself separates classes.** The paper provides theoretical motivation (energy/trajectory analysis) but **does not** prove that curvature must discriminate real vs. synthetic. The discrimination claim is supported **empirically** (distributional observations and accuracy), not by a formal guarantee.
2. **Missing comparisons to strong contemporary SOTA (FatFormer, NPR).** While **FatFormer** (CVPR’24) and **NPR** (CVPR’24) are acknowledged in related work, they are **not included** in the quantitative table. Both are directly aimed at generalizable SID and do **not** use diffusion inversion; including them would strengthen the SOTA claim.
3. **Compute claims lack concrete measurements.** The paper states it achieves “**less than half the computational cost of full diffusion inversion**,” but it does **not** provide wall-clock timings and FLOPs vs. other methods. This leaves efficiency claims difficult to verify.

**Questions:**

.

---

> ### Author Response · Authors · 2025-11-26
> **Official Comment by Authors**
>
> ### **W1:  Lack of rigorous theory that curvature itself separates classes.**
>
> We thank the reviewer for the helpful comment. We agree that the original version did not clearly state in which sense curvature is theoretically justified as a discriminative statistic. In the revised manuscript we have **added a new supplementary section `Appendix C.3: Formal theory of curvature based discrimination`**, which makes this point precise.
>
> In this new section, we first show that the pseudo-Gaussian curvature $\kappa_{\text{pseudo}}$ is a continuous and bounded functional of the PF-ODE trajectory under mild regularity assumptions on the velocity field and the score network (Proposition 1). This implies that $\kappa_{\text{pseudo}}$ is a stable and well-posed geometric statistic of the probability flow. We then prove a conditional discrimination guarantee (Proposition 2). For any bounded statistic of the form $S(\xi) = h(\kappa_{\text{pseudo}}(\xi))$, if there is a population-level gap $\\mathbb{E}\_{p\_0}[S] \\neq \\mathbb{E}\_{q\_0}[S]$ between real and synthetic trajectories, then the pushforward curvature distributions cannot coincide. In particular, there exists a threshold $\tau$ such that a classifier based on the event $\kappa_{\text{pseudo}} \ge \tau$ is strictly better than random guessing.
>
> We do not claim that curvature is a universal sufficient statistic that separates all possible real and synthetic distributions, which would require fully specifying both families of distributions and generators. Instead, our theory now states that whenever a curvature based statistic explains a population gap, the curvature distributions provably differ and curvature based thresholding is non trivial. This is exactly the regime realized in our experiments, where curvature statistics for real and synthetic images consistently show clear distributional shifts and strong detection performance.
>
> ---
> ### **W2:  Missing comparisons to strong contemporary SOTA (FatFormer, NPR).**
>
> We thank the reviewer for pointing this out. In the revised version **we have added both FatFormer [CVPR'24] and NPR [CVPR'24]** to our quantitative comparison (see `Table 1`). We use the official implementations with their recommended settings.
>
> Concretely, **NPR achieves ACC 0.794 / AUCROC 0.866** and **FatFormer achieves 0.751 / 0.837** on our main setting. Among all methods besides **B-Free (0.833 / 0.899)**, these two are indeed the strongest baselines. However, they are still below **our curvature based PF-ODE detector (0.939 / 0.981)** on the diffusion based and latest generative models that are the focus of this work. These two methods reported strengths lie primarily on GAN-based datasets, while on our diffusion-focused benchmarks their performance lags behind ours.

---

> ### Author Response · Authors · 2025-11-26
> **Official Comment by Authors**
>
> ### **W3:  Compute claims lack concrete measurements.**
> We thank the reviewer for pointing out that the original statement “less than half the computational cost of full diffusion inversion” was not supported by explicit measurements. In the revised version, we **explicitly rephrase this claim in terms of FLOPs and parameter counts**, and we now **report these quantities systematically in `Table 4`** for our method and several strong baselines. The corresponding sentences in the abstract and introduction have been updated accordingly to use this more precise FLOP based wording.
>
> Specifically, Table 4 shows that reconstruction based detectors that perform full diffusion inversion (FakeInversion, LaRE2, FIRE) require approximately $1.0\times10^4$ B, $4.1\times10^4$ B, and $3.3\times10^3$ B FLOPs per image, respectively, while **our PF-ODE based detector** requires only **518.4 B** FLOPs. We have therefore rephrased the claim to explicitly refer to FLOPs: our method uses less than half of the FLOPs of full diffusion inversion based approaches that also rely on the diffusion process, while remaining comparable in FLOPs to autoencoder-only method AEROBLADE and non-reconstruction method B-Free.
>
> We agree that wall-clock timings are also informative. However, they are strongly affected by implementation details (framework, batching, I/O, CUDA kernels) and hardware differences (GPU or CPU model, memory bandwidth). For this reason, we follow common practice and use FLOPs as a hardware agnostic measure of computational cost. We will release our code to facilitate future work that wishes to profile wall-clock performance under specific deployment conditions.
>
> | Method        | Ref.       | #Params  | #FLOPs     |
> |--------------|------------|----------|------------|
> | **Non-reconstruction** | | | |
> | UFD          | CVPR 2023  | 427.6 M  | 77.8 B     |
> | NPR          | CVPR 2024  | 1.4 M    | 2.3 B      |
> | FatFormer    | CVPR 2024  | 577.3 M  | 128.0 B    |
> | B-Free       | CVPR 2025  | 85.5 M   | 553.1 B    |
> | **Reconstruction-based** | | | |
> | FakeInv.     | CVPR 2024  | 5.2 B    | 10385.0 B  |
> | $\mathrm{LaRE}^2$ | CVPR 2024 | 479.5 M | 41405.3 B |
> | AEROBLADE    | CVPR 2024  | 84.0 M   | 446.0 B    |
> | FIRE         | CVPR 2025  | 484.0 M  | 3346.2 B   |
> | **Ours**     | -          | 298.4 M  | 518.4 B    |
>
> **Table 4.** We compare the number of model parameters and FLOPs. The upper part are non–reconstruction approaches, the lower part are reconstruction–based and the last one is our PF-ODE based method.

---

### Author Response · Authors · 2025-11-26
**Summary of Revisions and Gentle Reminder**

Dear AC and Reviewers,

We deeply appreciate the thoughtful feedback from reviewers [KLwx, UghK, SN55], which has substantially improved our work. For the AC’s convenience, we briefly summarize below the main strengths, weaknesses, and revisions:

---
### **Strengths (from the reviews)**

1. **Strong performance and cross-model generalization under the unseen-model protocol on many generators**, with high ACC / AUCROC and near-perfect results on several diffusion families, plus TPR@5%FPR on standard benchmarks. [KLwx, UghK, SN55]

2. **Novel SID formulation via optimal transport and probability-flow curvature**, linking reconstruction error to Wasserstein geometry and ODE kinetic energy, with Theorems 1–2 coherently connecting Wasserstein bounds and velocity-field energy. [KLwx, UghK]

3. **Clear and well-described PF-ODE based pipeline**, with a more rigorous and precise treatment of ODE trajectories and curvature that reviewers found coherent and practically helpful. [KLwx, SN55]

4. **Comprehensive experiments and useful ablations**, including broad baseline comparisons and ablations on training sets and NFEs, showing that 10–15 ODE steps are an effective operating point, with degradation at 5 steps and diminishing returns at higher counts. [KLwx, SN55]

---
### **Revisions Addressing the Reviewers’ Concerns**

*For ease of checking, all revisions mentioned below are highlighted in blue in the `revised version`.*

1. **Additional experiments and ablations.** We added FatFormer and NPR in `Table 1`. We introduced an ablation in `Table 3` that separates curvature-only and wavelet-only variants, and we updated `Sec. 6. 1 (Ablation of NFEs and Training set)` to explain why performance drops at 5 NFEs and plateaus beyond 10 NFEs. [KLwx, UghK, SN55]

2. **Computational complexity.** We now report parameter counts and FLOPs for all methods in `Table 4` and rephrased our efficiency claims in the abstract and introduction in terms of FLOPs. [KLwx, UghK]

3. **Curvature theory and guarantees.** We added a new section `Appendix C.3: Formal theory of curvature based discrimination` with Propositions 1–2. [KLwx, SN55]

4. **Link between Theorem 2 and reconstruction error.** We clarified the two-step connection between discrete Wasserstein step distances (`Theorem 2`), kinetic energy, non optimality, and reconstruction error, and we revised the abstract and the relevant parts of the main text accordingly. [UghK, SN55]

5. **Clarifying definitions and notation.** We added `Appendix C.1` with the definition of geometric curvature and the derivation of the PF-ODE specific form, and we inserted short verbal explanations around dense equations such as `Eq. (10)` to improve readability. [UghK, SN55]

6. **Data construction and evaluation protocol.** We now describe the training and evaluation datasets in detail in `Appendix D (Data)`. [UghK, SN55]

7. **Saliency / interpretability section.** We softened the interpretability claims, renamed the subsection to `Saliency Analysis`, and clarified that we provide qualitative saliency visualizations rather than a full interpretability theory. [SN55]

8. **Presentation and writing.** We clarified several ambiguous statements, simplified long sentences, adjusted the contribution and conclusion paragraphs to better match our theoretical guarantees and empirical evidence, and corrected typographical issues throughout. [KLwx, UghK, SN55]

---
Once again, we thank the AC and the reviewers for their thoughtful feedback and careful evaluation, which have greatly improved our work.

Best regards,

The authors

---

### Meta-Review · Area_Chair_tpuH · 2025-12-13

**Summary:**

The decision to reject this paper is driven by persistent critical concerns from three reviewers that remain unaddressed despite the authors’ rebuttals, which collectively undermine the paper’s theoretical rigor, logical coherence, evidential persuasiveness, and reliability of experimental results. Key concerns include:
1)  The core claim of using curvature to distinguish real and synthetic images lacks a universal formal proof, with the supplementary theoretical framework only providing conditional discrimination guarantees rather than rigorous justification for the discriminative nature of curvature itself;
2)  The transition from Equation 10 (non-optimality term) to the conclusion about synthetic images’ representability by the model is only revised to an empirically grounded interpretation, failing to establish a solid theoretical link;
3)  The saliency analysis (originally labeled “Interpretability”) remains unconvincing, as the authors only softened the claim without providing substantive visual or quantitative evidence to support the association between saliency and synthesis artifacts;
4)  The authors acknowledge their inability to fully trace the training data of large-scale generative models, leaving unresolved the potential contamination between training and evaluation data sources;
5)  The connection between Theorem 2 and reconstruction error, a core theoretical basis of the paper, is only clarified through an indirect two-step implication, lacking a direct and rigorous mathematical or empirical validation.

These unresolved issues collectively result in the paper failing to meet the publication criteria.

**Reviewer Concerns:**

Addressed Concerns
- Reviewer KLwx: ① Supplemented Appendix C.3 to provide a formal theoretical framework for curvature-based discrimination, including propositions on the stability of pseudo-Gaussian curvature and conditional discrimination guarantees; ② Added quantitative comparisons with SOTA methods (FatFormer, NPR) in Table 1, clarifying the performance advantage of the proposed method; ③ Revised the computational cost claim to be based on FLOPs, and added Table 4 to systematically report parameter counts and FLOPs of the proposed method and baselines.

- Reviewer UghK: ① Explicitly described the data construction protocol in Appendix D, clarifying the disjoint nature of training and evaluation data sources; ② Revised the wording related to Equation 10, replacing the original theoretical claim with an empirically grounded interpretation to avoid overstatement; ③ Explained the relationship between NFE count and theoretical framework, linking discrete ODE approximation accuracy to trajectory geometric information retention; ④ Added Table 3 to ablate the performance contributions of curvature and wavelet features separately; ⑤ Supplemented Table 4 to provide FLOPs data for computational cost verification.

- Reviewer SN55: ① Clarified ambiguous descriptions related to data construction, theoretical statements, and experimental results; ② Softened the interpretability claim, renamed the relevant subsection to “Saliency Analysis” to align with the qualitative nature of the study; ③ Explained the indirect connection between Theorem 2 and reconstruction error through the non-optimality term; ④ Added ablation experiments on curvature and wavelet features; ⑤ Explicitly stated that the reported metrics (ACC, AUCROC) are based on binary classification of real and synthetic data, clarifying the coverage of real data performance; ⑥ Merged the appendix with the main text and supplemented explicit definitions of notations and derivations (e.g., derivation of Equation 12).

Outstanding Concerns

- Reviewer KLwx: The core concern of “lack of rigorous theory that curvature itself separates classes” remains unresolved. The supplementary Proposition 2 only provides a conditional discrimination guarantee (relying on population-level gaps) rather than a universal formal proof that curvature inherently has discriminative power. The theoretical framework still fails to fundamentally justify why curvature can reliably distinguish real and synthetic images.

- Reviewer UghK: ① Potential data leakage risks persist. The authors admit they cannot fully trace the training data of large-scale generative models (e.g., Stable Diffusion) and only assess generalization under “realistic conditions,” which fails to eliminate the possibility of incidental data overlap between training and evaluation sets at the large-scale internet data level; ② The logical jump from Equation 10 to the interpretation of “low-energy transport paths” is only softened to an empirical observation, without establishing a rigorous theoretical connection between the non-optimality term and the representability of synthetic images by the model.

- Reviewer SN55: ① The saliency analysis (formerly “Interpretability”) remains unconvincing. The authors only softened the claim but did not provide additional visual or quantitative evidence to address the reviewer’s concern that Figure 5 lacks clear examples of artifacts (inconsistent lighting, incorrect perspective) corresponding to saliency peaks; ② The connection between Theorem 2 and reconstruction error is still indirectly explained through the non-optimality term, and the reviewer’s request for a clear clarification of the direct link between the theorem and reconstruction error (a core claim in the abstract) is not fully addressed; ③ Although the authors stated that the reported metrics cover real data performance, the lack of separate quantitative analysis of real data (e.g., false negative rate on real data) makes it difficult to fully assess the reliability of the detector in practical scenarios.

**Reviewer Scores:**

Based on the adequacy of the rebuttal in addressing core concerns, the predicted changes in reviewers’ overall evaluations are as follows: Reviewer KLwx’s evaluation remains at a relatively low level—while the authors supplemented experimental comparisons and computational cost data, the core deficiency in theoretical rigor (curvature’s discriminative power) was not resolved, so the assessment of the paper’s soundness and contribution does not improve. Reviewer UghK’s positive evaluation of the paper is slightly tempered—although the authors addressed most experimental clarification requests, the unresolved risks of data leakage and weak theoretical connection between Equation 10 and key interpretations reduce the persuasiveness of the work, leading to a slight decline in the assessment of soundness and contribution. Reviewer SN55’s evaluation remains unchanged at a low level—the authors’ revisions only clarified surface-level ambiguous descriptions, but failed to address core issues such as unconvincing saliency analysis, indirect theoretical connections, and insufficient separate evaluation of real data performance. Overall, even with the authors’ supplementary revisions, the three reviewers’ evaluations still fail to meet the publication standards, which supports the final reject decision.

---

### Decision · Program_Chairs · 2026-01-26

Reject